# High-speed 3D DNA PAINT and unsupervised clustering for unlocking 3D DNA origami cryptography

Gde Bimananda Mahardika Wisna [1,2,3] ✉, Daria Sukhareva [2,4], Jonathan Zhao [2,5], Prathamesh Chopade [2], Deeksha Satyabola [2,4], Michael Matthies [2], Subhajit Roy [1,2,3], Chao Wang [2,6], Petr Šulc [2,3,4], Hao Yan [2,4] & Rizal F. Hariadi [1,2,3] ✉

DNA origami information storage is a promising alternative to silicon-based data storage, offering a molecular cryptography technique concealing information within DNA origami. Routing, sliding, and interlacing staple strands lead to a large 700-bit key size. Practical DNA data storage requires high information density, robust security, and accurate and rapid information retrieval. Consequently, advanced readout techniques and large encryption key sizes are essential. Here, we report an enhanced DNA origami cryptography protocol in 2D and 3D DNA origami, increasing the encryption key size. We employ all-DNA-based steganography with fast readout through high-speed DNA-PAINT super-resolution imaging. By combining DNA-PAINT data with unsupervised clustering, we achieve an accuracy of up to 89%, despite the flexibility in the 3D DNA origami shown by oxDNA simulation. Furthermore, we propose criteria that ensure complete information retrieval for the DNA origami cryptography. Our findings show that DNA-based cryptography is a secure and versatile solution for storing information.

The development of semiconductor-based transistors in the mid-20th century marked the beginning of the information revolution, enabling data storage and computing, while the advent of fiber optics and optical amplifiers necessitated secure communication systems[1]. Modern cryptography, such as the Advanced Encryption System (AES) with keys up to 256 bits, evolved to meet this need, relying heavily on semiconductor devices for processing power[2–4]. However, the limited availability of these semiconductor devices, due to high demands in many advanced technologies with long lead time to manufacture[5], and the high energy demand for operating devices[6] present significant challenges. DNA, with its stability, programmability, high information density, and low maintenance needs, has emerged as a promising substitute[7–10]. Pioneering work on DNA computing[11–17] and archival data storage[18–21] has demonstrated DNA's potential for these applications, making the development of molecular cryptography protocols crucial for secure DNA-based archival data storage management in the future.

Recent advancements in molecular cryptography for 2-way communication have been demonstrated through a multitude of chemical approaches, such as harnessing optical and physical properties of molecules[22–25] and exploiting Watson-Crick base pairing of DNA[26–30]. Zhang et al. developed DNA origami cryptography, which relies on the challenge of accurately predicting the arrangement and interlacing of DNA strands to achieve determined geometric shapes that function as templates for encrypting patterns[31]. This approach, similar to steganography, involves folding a long DNA strand from the M13mp18 bacteriophage with hundreds of short single-stranded staple strands,

[1]Department of Physics, Arizona State University, Tempe, AZ, USA. [2]Center for Molecular Design and Biomimetics at the Biodesign Institute, Arizona State University, Tempe, AZ, USA. [3]Center for Biological Physics, Arizona State University, Tempe, AZ, USA. [4]School of Molecular Sciences, Arizona State University, Tempe, AZ, USA. [5]School of Computing and Augmented Intelligence, Arizona State University, Tempe, AZ, USA. [6]School of Electrical, Computer and Energy Engineering, Arizona State University, Tempe, AZ, USA. ✉e-mail: gwisna@asu.edu; rhariadi@asu.edu

each <100 nt in length[32–34]. The central principle of modern cryptography, which employs complex mathematical problems to generate large possibilities of keys[2], can be seamlessly translated into the DNA-origami system[28]. Zhang et al. estimated with some mathematical simplification that the key space resulted from self-avoiding scaffold foldings can exceed 700 bits with a DNA scaffold of 7249 nucleotides forming 2D DNA origami, as found in the M13mp18, potentially surpassing the Advanced Encryption Standard (AES) by a factor of >2[31]. However, enumeration of self-avoiding polygon according to perimeter or area is actually still an unsolved problem, indicating an astronomically large number of folding paths which goes beyond 700 bits[35]. This will provide an opportunity to have a bigger key size to have more secure encryption if we can utilize the folding paths to form not only 2D but also 3D DNA origami.

Despite the theoretical large key space, the practicality of the current demonstration of DNA origami is limited by the inherently slow imaging of conventional atomic force microscopy (AFM)[36] coupled with the reliance on biotin-streptavidin conjugations, which frequently result in undesired aggregations[37,38]. Another way to read data stored in mostly tubular-shaped DNA duplex decorated with DNA nanostructures is by utilizing nanopore sequencing[39,40]. Chen et al. and Zhu et al. have shown that multilevel encoding with different barcodes with DNA nanostructure multi-way junctions and dumbell shapes can improve the data readout using nanopore[41,42]. It can reach readout resolution of different nanostructure barcodes with separation up to 6 nm within tens of microsecond for one translocation event of duplex with length of around 100–200 nm[41]. However, due to the geometry of the nanopore and the readout process of the data which has to be in a sequential manner as the samples translocate the pore, the application for DNA origami data storage readout is limited to DNA nanostructures with tubular form. On the other hand, high-speed DNA points accumulation for imaging in nanoscale topography (high-speed DNA-PAINT)[43], a DNA-based super-resolution imaging technique, offers a solution to enhance readout speed and eliminate aggregation induced by protein conjugation. This method capitalizes on the stochastic binding of single-stranded DNA (ssDNA), referred to as docking strand, and short fluorophore-labeled ssDNA called imager strands to achieve resolutions as high as sub-nm[44] to 10 nm[45]. DNA-PAINT enables faster imaging and a larger imaging area of nearly 100 μm by 100 μm, allowing for the imaging of thousands of complex shapes of 1D, 2D, and 3D DNA origami nanostructures per field of view[43]. Despite the application of DNA origami and DNA-PAINT techniques for alternative DNA storage, high error rates during experimental origami folding and low detection efficiency in DNA-PAINT imaging necessitate the development of error-correction post-processing algorithms[21]. Therefore, optimizing these strategies is crucial for improving information retrieval in DNA-based systems.

In this work, we report the application of high-speed DNA-PAINT and unsupervised clustering to achieve DNA origami cryptography in 2D and 3D DNA origami with fast readout while maintaining high detection efficiency of docking strands bearing pattern information. By extending the binding length to the scaffold, we achieve a detection efficiency of ~90% using DNA-PAINT. The process allows quick analysis of thousands of DNA origami, each with unique patterns, within ~24 min at a resolution of 10 nm. Combining high-speed DNA-PAINT readout with unsupervised k-means clustering enables fast and accurate assignment of centroids to 2D or 3D clusters and template alignment based on least-squares distance minimization with pattern matching. Integration of bit redundancy improves the accuracy in the retrieval of data from 2D and 3D DNA origami encryption and decryption processes, achieving an accuracy of 70–89%, despite the inherent flexibility of 3D DNA origami architectures, as predicted by computational modeling of oxDNA[46,47]. The 3D alignment of the 3D DNA-PAINT data to the mean structures as simulated by oxDNA gives better root mean square deviation (RMSD) compared to fitting the 3D DNA-PAINT data to the unrelaxed structures. We also demonstrate an encryption using dimer assembly from two rigid 3D DNA origami monomers, achieving agreement between the designed pattern and 3D DNA-PAINT imaging. Our method streamlines DNA origami cryptography, with the advantage of many possible folding paths, using fast DNA-PAINT and unsupervised clustering to achieve secure information transmission with high readout accuracy.

## Results

### The DNA origami cryptography protocols

The 2-way DNA origami cryptography protocol uses symmetric keys between a sender, Alice, and a receiver, Bob. Alice aims to securely transmit the message ASU (Fig. 1A). Using the principles of symmetric cryptography, Alice encrypts the message and generates keys and cipher text, which we refer to as a cipher mixture comprising multiple DNA strands. The encryption and the decryption processes involve 3 steps: encoding the letters and their corresponding positions as a binary pattern on DNA origami (🔒 1), docking sequence assignment (🔒 2), and specifying the shape and folding pathway of the scaffold strand (🔒 3). Alice and Bob can exchange the cipher mixtures, consisting of M13mp18 scaffolds and docking strands that encode the entire letters and positions of the message, through insecure communication channels. However, they must securely exchange the 3 keys required for decryption. To decrypt the message, Bob reverses the encryption process by first folding the DNA origami (🔓 3) using the staple strand mix (key 3), performing high-speed DNA-PAINT imaging to reveal the pattern (🔓 2) using the imager strand (key 2), and then applying clustering and template alignment to extract the letters and their corresponding positions (🔓 1) based on the pattern rules (key 1). In the absence of any of the required encryption keys, potential adversaries will be incapable of decrypting the encoded data, thereby ensuring the security of the information (Fig. 1B, C).

### DNA origami encryption

The message encryption employed the DNA origami technique, involving uniquely designed routing of a long M13mp18 scaffold and staple strands for DNA origami geometries, such as a 2D Rothemund Rectangular Origami (RRO), or a 3D wireframe cuboctahedron origami (Fig. 1B; Supplementary Tables 1 and 2). The encryption protocol (Fig. 1A) flows downward (red arrows) resulting in the creation of 3 keys. The process began by converting the text message (Fig. 1A) into binary codes (Supplementary Table 3), that denoted each letter and its position within the text (Fig. 1A). Next, a geometric shape of DNA origami was selected to serve as a template for the pattern, and a pattern rule that the template can accommodate was chosen. In our first demonstration, we selected the 2D RRO with dimensions of approximately $90 \times 60$ nm$^2$ as the template and devised a pattern encryption rule (Fig. 1A) allowing the arrangement of the binary code pattern, that consisted of 0 and 1 bits, on the RRO template (Fig. 1A). The patterns on the DNA origami can achieve a resolution of sub-100 nm, with a separation of 10–20 nm between two bits in a pattern. The alignment markers (gray circles in Fig. 1A) are necessary to break the rotational symmetry within the plane of the RRO.

The second step of encryption required designing a unique docking strand sequence for high-speed DNA-PAINT imaging, where we adopted the sequence used by Strauss et al.[43] (Fig. 1A); this step was called docking assignment. The third and final step was to generate the staple strands that will fold the M13mp18 scaffold strand into an RRO and to assign docking sequences to extend specific staple strands on the RRO DNA origami that are designated for the 1 bits in our binary codes (Fig. 1A). These extended strands are interchangeably called information strands or docking strands throughout the article. The strands on an RRO corresponding to 0s would have no docking sequence extension and are called staple strands key. The possibility space for the generation of the set of staples and the routing of the

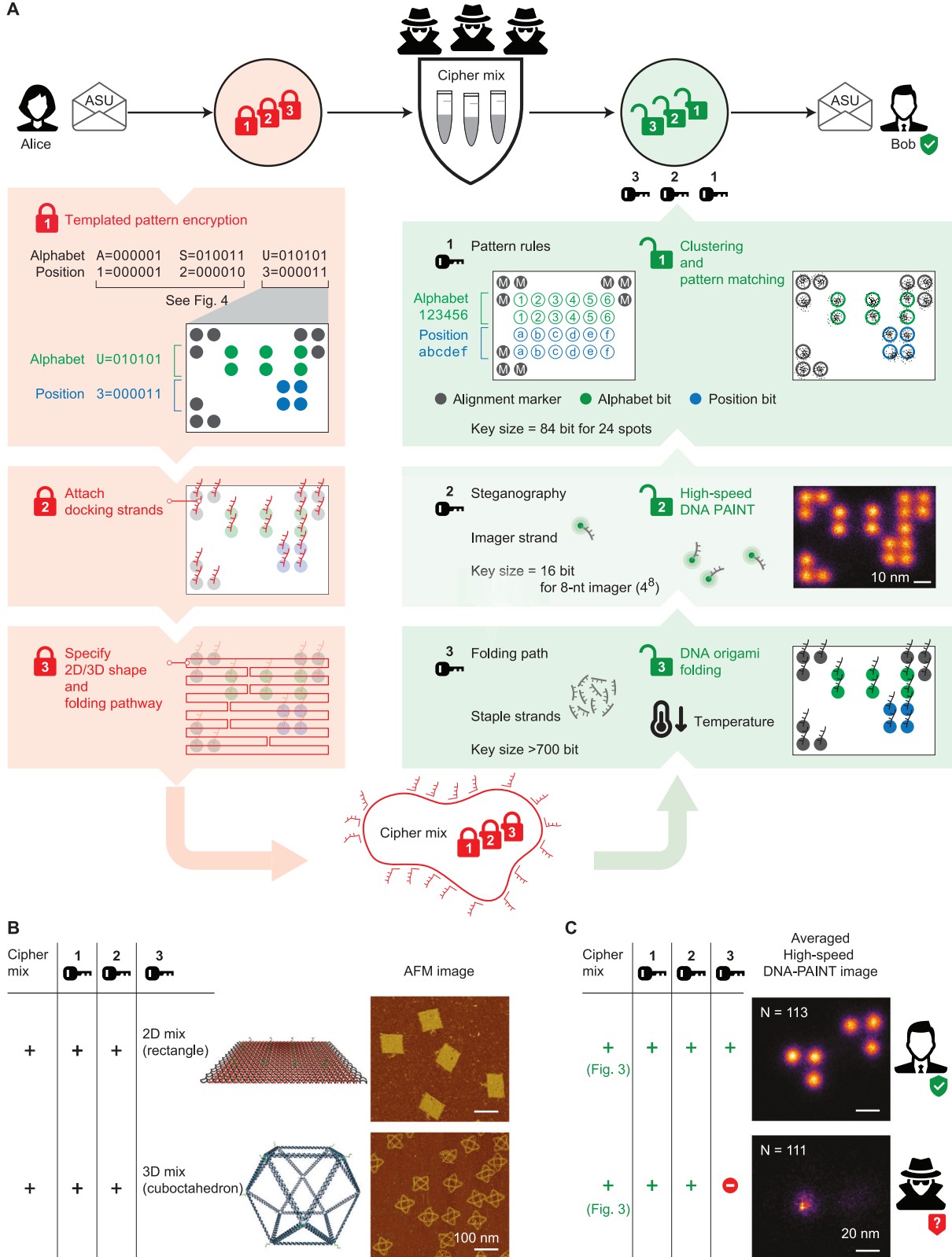

**Fig. 1 | DNA origami cryptography protocol. A** 3-step encryption/decryption process of information encoded in DNA strand cipher mix. Alice encrypts a message (ASU) through templated pattern encryption, docking strand placement, and DNA origami folding. Bob decrypts using DNA origami formation through annealing, high-speed DNA-PAINT imaging, clustering, and pattern matching. **B** 2D and 3D DNA structures for DNA origami cryptography and their corresponding representative AFM images. **C** Averaged DNA-PAINT images demonstrate successful decryption with all keys (top) and failed decryption with 1 missing key (bottom).

scaffold was estimated to be >700 bits for M13mp18, ensuring the security of information transfer[31]. Finally, the cipher mixture was generated by mixing the scaffold and the information strands. In this demonstration, we have 3 cipher mixtures since we have 3 letters: A at position 1, S at position 2, and U at position 3. Each cipher mixture consisted of universal M13mp18 scaffold strands with all corresponding information strands for each letter and position (Fig. 1A; Supplementary Table 4). The encryption process had 3 keys: the pattern encryption rules (key 1), the docking sequence (key 2), and the set of staple strands (key 3). The pattern and docking encryption added extra layers of security. In our first demonstration for 24 pattern spots, the key size for the pattern encryption was 84 bits[31]; a larger number of spots resulting in a larger key size. For the docking encryption, an 8-nt length docking sequence with 4 possible nucleotides at each position resulted in $4^8$ possible combinations, equivalent to a 16-bit key. However, these additional layers of security were not the primary security measures, as the total key sizes for these two steps were significantly smaller than the key size of the DNA origami encryption. In the case where the adversaries tried to use spurious interaction to image the docking, it would only result in blurry pattern due to short binding of spurious interaction or even if they could obtain the exact docking sequence through DNA sequencing, the true pattern would not be revealed via DNA-PAINT super-resolution imaging because of the highly secured DNA origami encryption which is impossible to crack[35] (Fig. 1C).

## The DNA origami decryption and readout

The decryption process involved the sequential application of 3 keys to the cipher mixtures (Fig. 1A, green boxes). First, the DNA staple strands key was applied to the mixtures, forming the RRO DNA origami through appropriate annealing (Fig. 1A). Next, DNA-PAINT super-resolution imaging was performed by applying the reverse complementary sequence of the docking strand, labeled with a Cy3B fluorophore, as the imager strands. High-speed DNA-PAINT docking enabled the acquisition of a super-resolution image containing thousands of origami within a single 12-min, 15,000-frame field of view. Averaged images of hundreds of DNA-PAINT images for each letter (Fig. 1C(top)) revealed the exact design and encrypted pattern (compared with Fig. 3A). Without the staple strand key, DNA-PAINT yielded random patterns, effectively concealing the true pattern (Fig. 1C(bottom)).

High-speed DNA-PAINT super-resolution imaging offers advantages over AFM due to its localization-based data, which seamlessly integrate with many clustering methods such as supervised machine learning[48,49], Bayesian approaches[50], and unsupervised machine learning for clustering[51]. In our implementation, we employed k-means clustering to obtain the centroids of each cluster in a pattern and used them for template alignment. The encoded binary was extracted by comparing the alignment with the pattern encryption rules to retrieve the ASU information. This process highlighted the importance of the 3 keys in the decryption process. Without the staple strands key, the true pattern remains concealed (Fig. 1C(bottom)). In the absence of the docking sequence knowledge, DNA-PAINT cannot be performed to reveal the pattern, and attempting AFM imaging of the pattern formed by only 19-nt ssDNA docking sequence extensions is not practically feasible, as demonstrated by the invisibility of 12 docking extensions in the case of RRO and cuboctahedron (Fig. 1B and Supplementary Fig. 1). The pattern encryption rule key is crucial for translating the revealed pattern into binary codes to retrieve the encrypted information.

## Detection and incorporation efficiency of information strands on 2D DNA origami via DNA-PAINT

Ensuring secure information transmission requires a high level of information retention and retrieval. Our encryption-decryption protocol relies on DNA origami architecture with integrated DNA-PAINT and unsupervised clustering readout. Previous work on DNA origami combined with DNA-PAINT readout for storage applications[21] showed a high error rate, thus requiring a mechanism and algorithm for robust error correction for each origami data droplet. Therefore, it is crucial to assess the incorporation and detection efficiency of the information strands on the RRO origami template to achieve higher information retention and retrieval. The two metrics are the measures on the ability to incorporate the docking strands on the origami and to be able to image and visualize them with high-speed DNA-PAINT. We investigated the detection efficiency of information strands on the 2D RRO DNA origami template via DNA-PAINT, where 1 information strand would be translated into 1 docking spot. To this end, we varied the number of spots per origami between 12, 24, and 48 as shown in Fig. 2A and Supplementary Fig. 2. The increasing number of spots per origami led to a reduction in the separation between spots. Specifically, the shortest distance between spots in origami with 12, 24, and 48 spots were systematically designed to be 20, 14, and 10 nm, respectively.

## Extending the length of the information strand binder on DNA origami leads to higher incorporation and detection efficiency

We started with an RRO DNA origami containing 12 spots/origami, with a 32-nt information strand binder (Fig. 2A(i)), and estimate detection efficiency following the method developed by Strauss et al.[52] (Supplementary Fig. 3). Information strand binder was a section in the information strands attached to the origami scaffold, thus facilitating placement of the information strand on a specific site on the origami (Fig. 2A, green-colored section). To improve the incorporation efficiency, we increased the information strand binder to 64-nt (Fig. 2A (ii–iv)), a strategy that has been shown to increase the melting temperature of the staple-scaffold complex during the annealing process[53]. We hypothesized that extending the information strand binders enhanced the stability of the information strand attachment on the DNA origami, notwithstanding physical perturbations arising from PEG purification, freeze-thaw cycles, pipetting, mixing, and local joule heating incurred during high-intensity DNA-PAINT imaging, thereby enhancing incorporation and detection efficiency.

Increasing the information strand binder length from 32-nt to 64-nt improved the probability of origami with all 12 dockings, while the detection efficiency varied depending on the docking location on 2D DNA origami. Fig. 2B–E presents the results for 12 spots per origami with varying binder lengths and numbers of information strands. Each row displays samples of DNA-PAINT from 4 individual origamis (Fig. 2B), an averaged image from N origamis or particles (Fig. 2C), the detection efficiency of each docking position (Fig. 2D), and the distribution of detected dockings (Fig. 2E). The probability of origami with all 12 dockings increased from 0.25 to 0.4 when the binder length was increased from 32-nt to 64-nt, despite a negligible increase in the mean detection efficiencies from (88.8% ± 10.0%) to (89.1 ± 12.4%) (Fig. 2C–E). Using the offset of 7% established by Strauss et al.[52], the mean incorporation efficiencies were (95.8 ± 10.0%) and (96.1 ± 12.4%) for the 32-nt and 64-nt binders, respectively. These findings suggest that longer binders increase the likelihood of information strands remaining bound to the origami. The detection efficiency was highest (>94%) in the center areas of the origami and lowest at the corners and edges (Fig. 2C, D), consistent with the findings of Strauss et al.[52].

By increasing the density of docking spots per origami with a 64-nt binder the detection efficiency was affected while necessitating the optimization of DNA-PAINT imaging parameters. We investigated the effect of higher docking density on detection efficiency by increasing the number of docking spots to 24 and 48 per origami (Fig. 2(iii and iv)). Consistent with the binomial distribution, the highest probability in the distribution did not correspond to full docking detection in both cases. The mean detection efficiencies were (87.92 ± 7.1%) and (89.16 ± 5.6%) for 24 and 48 docking spots, respectively, translating to

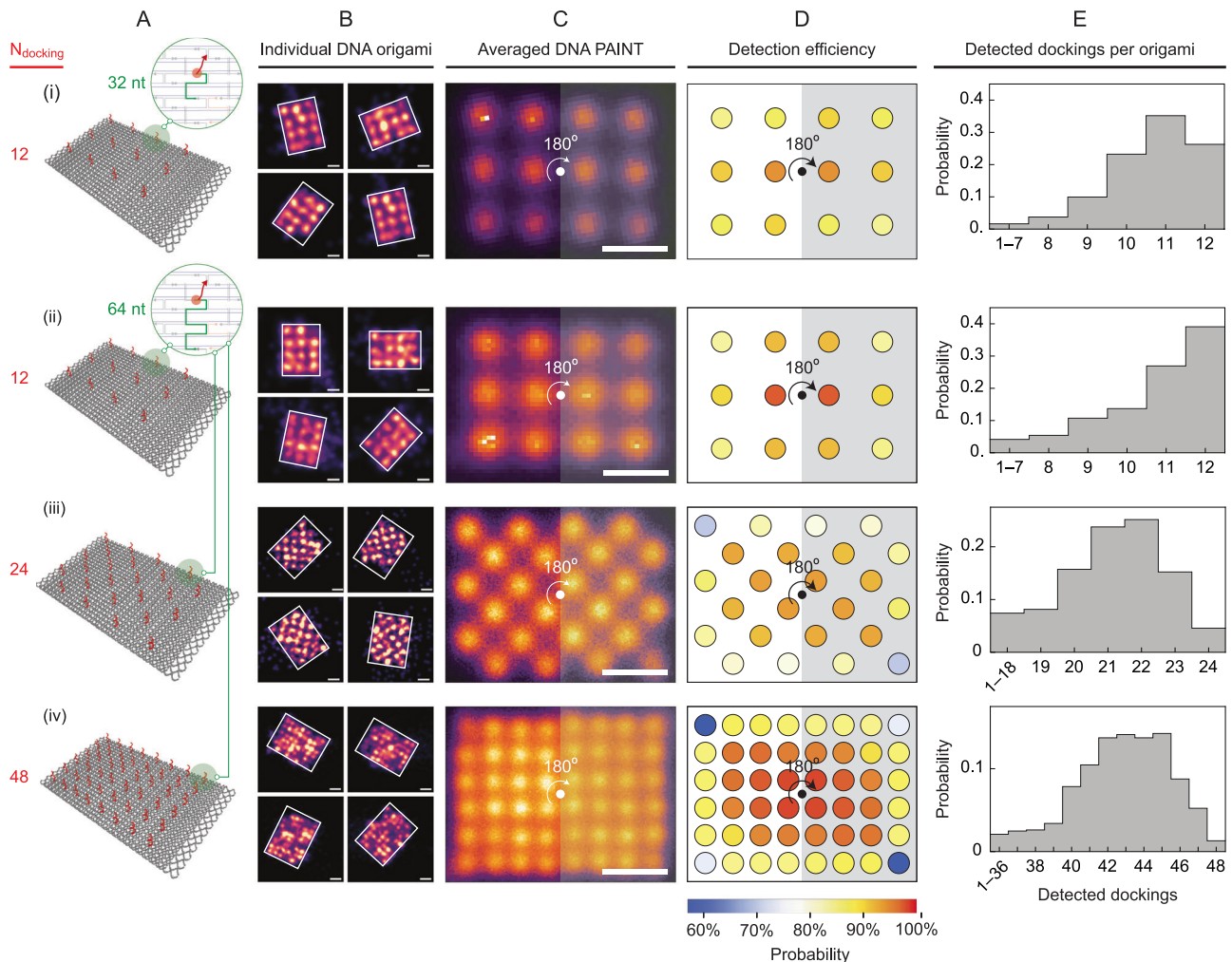

**Fig. 2 | Detection efficiency of information strands on 2D DNA origami.**
**A** Schematics of 2D RRO with red and green-colored docking strands: (i) 12 strands with 32-nt binders, (ii) 12 strands with 64-nt binders, (iii) 24 strands with 64-nt binders, and (iv) 48 strands with 64-nt binders where the binder is colored with green. **B**–**E** Results for 2D RRO with varying numbers and lengths of symmetrically-placed information strands. **B** Examples of 4 individual DNA-PAINT origami of the denoted system. **C** Averaged DNA-PAINT images (N = 3500, 5993, 1793, and 765 DNA origami, top to bottom, respectively), scale bars are 20 nm. **D, E** Detection efficiency of each docking position (**D**) and the number of detected docking (**E**). Due to rotational symmetry, pairs of symmetric docking positions have identical averaged probabilities.

incorporation efficiencies of (94.92 ± 7.1%) and (96.16 ± 5.6%). These results suggest that detection efficiency was influenced by both docking strand incorporation and the combined effects of docking density and DNA-PAINT imaging. Increasing docking density posed challenges for DNA-PAINT imaging, as closer spatial separations of docking spots (20 nm, 14 nm, and 10 nm for 12, 24, and 48 spots, respectively) needed adjustments in the imager strand concentration and acquisition frames to fully resolve the dockings (Supplementary Table 5). Despite these adjustments, high-speed DNA-PAINT allowed for reasonably fast imaging times of 12 min and 75 min for 12 and 48 docking spots, respectively, which is 4–15 times faster than AFM imaging for pattern readout on hundreds to thousands of origami per field of view.

The detection efficiency of docking spots via DNA-PAINT exhibited a systematic gradient across RRO, with its maximum value near the center and progressively lower values toward the edges and corners of RRO. However, increasing the docking density allowed for more dockings with high detection values. The center locations of the origami had consistent detection efficiencies >90%, while the edge and corner dockings had lower values (Fig. 2D). A higher detection efficiency ensured high information retention and retrieval for our

DNA origami encryption protocol. Despite the constraint imposed by our strategy of using longer binding to the scaffold for the information/docking strand limits the density to 48 dockings/origami (corresponding to 10 nm resolution), this resolution allowed for the theoretical encryption of $2^{28} \approx 268.4$ million combinations of numbers, letters, and punctuation marks that form texts and paragraphs, assuming near 100% incorporation and detection of information/docking strands (Supplementary Fig. 4).

## The DNA-PAINT and unsupervised clustering readout
We successfully executed the encryption and decryption of the text NSF using DNA origami cryptography, showcasing the robustness and accuracy of our encryption and readout protocols (Fig. 3). The pattern encryption rules could accommodate $2^3 = 8$ combinations of letters, numbers, and punctuation marks; however, in this demonstration, we encoded merely a 3-letter text of NSF. Using a 32-nt binder for the docking/information strands, DNA-PAINT imaging was performed for 12 min, yielding several hundred to a thousand super-resolution images of each pattern (Supplementary Table 5). Around 100 particles of each pattern were selected for readout accuracy analysis (Fig. 3A and Supplementary Fig. 5). K-means clustering was employed to obtain

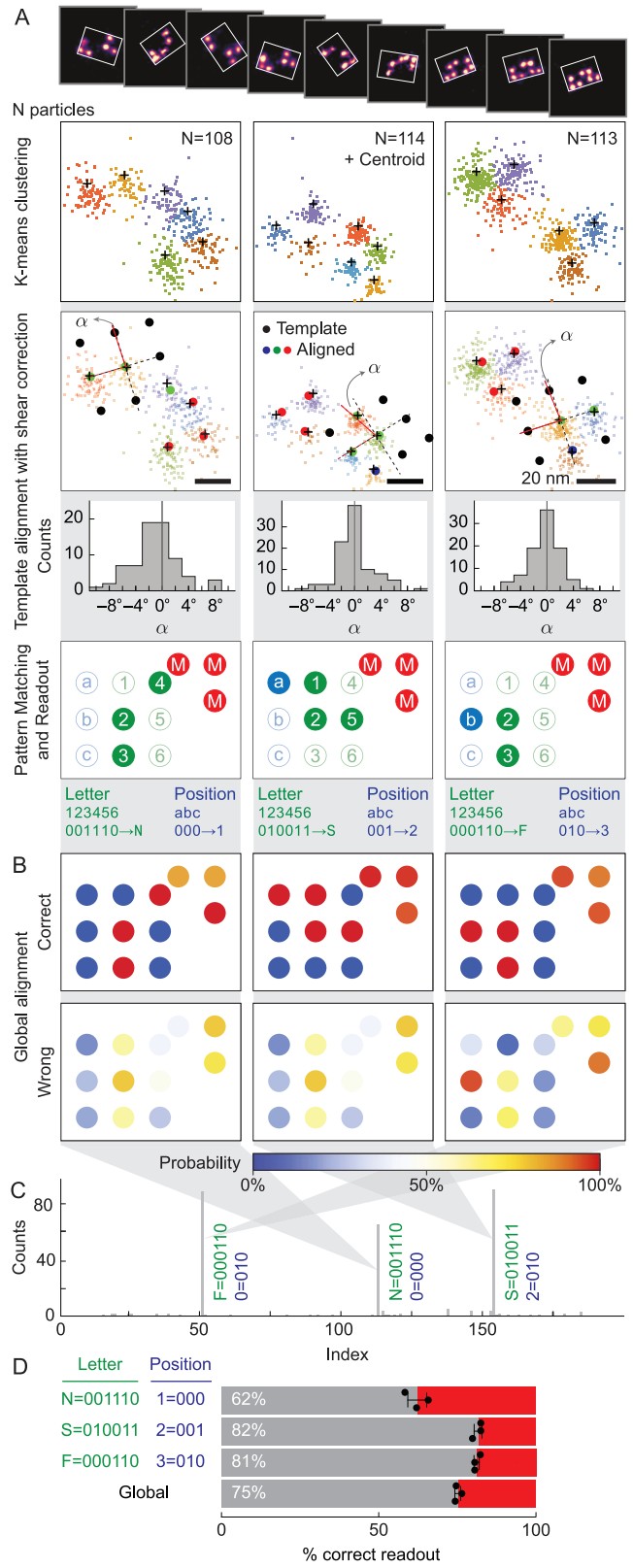

**Fig. 3 | Unsupervised k-means clustering and template alignment for NSF data readout. A** Workflow of the process: *N* particles, k-means clustering, template alignment with shear correction and distribution of shear angle α in degrees for each correctly read letter, and pattern matching and readout, scale bars are 20 nm. **B** Global bits alignment of correct (top) and wrong (bottom) readouts, displaying incorporation efficiency for each bit position using the method described in (**A**). **C** NSF data readout result presented as letter index vs. counts, showing 3 correct-readout major peaks and wrong-readout minor peaks. **D** Correct (gray) and wrong (red) readout percentage for each letter. Data are presented as mean values ± SD with *N* = 3 runs of the process in (**A**) using the same dataset with *N* = 108 for N letter, 114 for S letter, and 116 for F letter.

during the immobilization process on the glass coverslip. Bit extraction and readout were executed following the template alignment process.

The analysis of individual letter accuracies and the global accuracy of the NSF text demonstrated the robustness and suitability of the DNA origami encryption and readout protocols for secure communication between two parties. Global bit alignment analysis (Fig. 3B) revealed 100% detection efficiency for non-alignment marker dockings in correct readouts. In contrast, incorrect readouts showed undetected information-bearing dockings and non-zero detection in locations where no information dockings should be present, indicating misalignment of some origami. The combined readout results depicted in (Fig. 3C) have prominent peaks corresponding to the N,1, S,2, and F,3 letters, with wrong readouts occupying many small peaks. The count ratio between the lowest main peak and the highest false readout peak was 13.4, whereas the ratio between the highest main peak and the highest false readout peak was 18.4. Figure 3D displayed the individual letter accuracies with error bars denoting the standard deviation from three different runs of k-means clustering, template alignment by cost function minimization, and bit extraction. Despite the non-deterministic nature of the minimization, each letter has reasonable accuracy (>62%) with a relatively small error of ≤3%, indicating the method's robustness. The global accuracy of the NSF text is ~75% with a relatively small error of 0.9%.

**Incorporating bit redundancy in the encryption rules improves the readout accuracy of higher information density DNA origami with 48 docking spots per origami**

Following the successful demonstration of 3-letter DNA origami encryption and readout with 20 nm spaced docking spots, we designed encryption rules to increase the information density using 48 docking spots per origami. Based on the detection efficiency results (Fig. 2(iv)), 3 spots from each of the 4 corners were assigned as orientation markers, providing 3 redundancies to compensate for the lower detection and incorporation efficiency of corner dockings as opposed to other docking sites (Fig. 4A). 6 bits were allocated for encrypting letters, numbers, and punctuation marks through binary representation, alongside an additional 6 bits for position/order, accommodating up to $2^6$ = 64 characters to form a text. A total of 12 bits were mapped onto 12 docking spots on the origami, with 1 additional redundancy for each bit (highlighted in red in Fig. 4A). This scheme required 24 docking spots in total (Fig. 4), with green circles representing the character bits and blue circles indicating the position bits.

The effect of redundancy on readout accuracy is investigated by encrypting the ASU text (A,1, S,2, and U,3 in Fig. 4). We selected 200 particles of each pattern for readout accuracy analysis, with averaged images shown in Fig. 4B (see Supplementary Fig. 6 for all analyzed picks). DNA-PAINT imaging takes over a duration of 24 min (30,000 frames), generating datasets ranging from hundreds to thousands of super-resolution images per pattern (Supplementary Table 5). K-means clustering and template alignment were performed. The

centroids corresponding to each docking (Fig. 3A), with the optimal number of clusters determined by the elbow method (see "Methods" for details). Template alignment was then used to align the centroids to the pattern encryption rules, accommodating origami with shear angles α (Fig. 3A). Although the majority of DNA origami structures exhibit minimal shear angles nearing 0°, some correctly aligned origami experience significant shear deviations, presumably induced

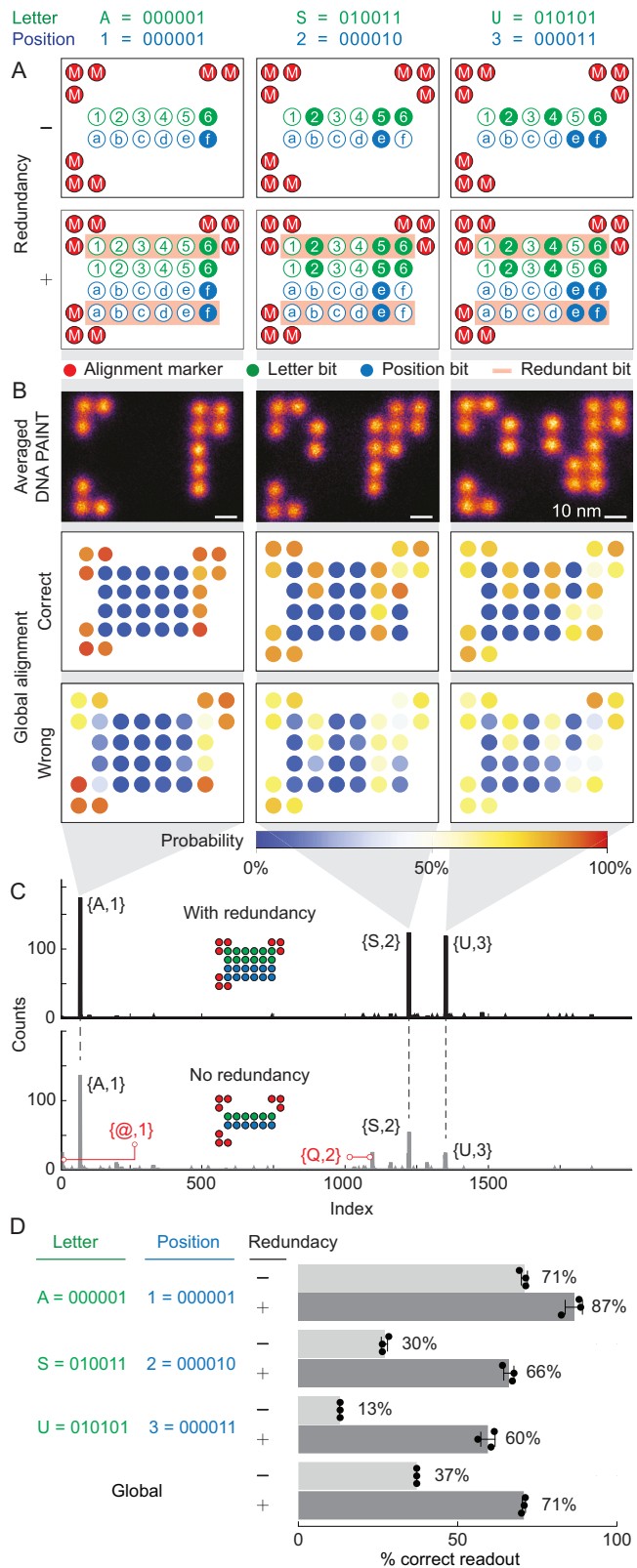

**Fig. 4 | Bit redundancy on higher density 2D RRO. A** Pattern encryption rules for ASU dataset showing alignment marker, letter bits, position bits, and redundancy. **B** Summed DNA-PAINT images of 3 letters of ASU (top), ($N = 200$ for each A, S, and U letter), scale bars are 10 nm, and global bits are aligned with the correct readout (middle) and wrong readout (bottom), showing incorporation efficiency for each bit position. **C** Readout of the ASU dataset presented as letter index vs. counts, analyzed with redundancy (top) and without redundancy (bottom). **D** Readout percentage of correct and wrong readout for each letter and global; Data are presented as mean values ± SD with $N = 3$ runs of readout process using the same dataset.

3 letter patterns was compared with and without bit redundancy (Fig. 4C, D). With redundancy, 3 clear main peaks corresponding to the correct letters and positions were observed, along with small peaks of wrong readout (Fig. 4C(top)). The ratio between the lowest main peak and the highest false readout peak was 13.2, and 19.3 between the highest main peak and the highest false readout peak. In contrast, without redundancy, the combined readout failed to retrieve the encrypted information, showing comparable false readout peaks of @,1 and Q,2 that interfere with the final readout (Fig. 4C(bottom)).

Adding one-bit of redundancy significantly improved the accuracy of each letter and the global accuracy in higher docking density encryption. Figure 4D showed the accuracy for each letter with and without bit redundancy. The global accuracy improved from ~37% to ~71% by adding one redundancy. The small standard deviation (<3%) from 3 different runs of k-means clustering, template alignment, and bit extraction demonstrated the robustness of our readout method. Two redundancies, utilizing all 48 docking spots, do not yield significantly better results than 1 redundancy (Supplementary Figs. 7 and 8), indicating that the maximum global readout accuracy achievable by redundancy for these 3 letters (ASU) is around 70–75%. False readouts can arise from factors such as imperfect imaging, causing the closest 2 spots to be poorly resolved, leading to failures in k-means centroid assignment and template alignment. We observed that the accuracy decreases as the number of bits occupied by a specific letter increases, as evident when comparing A,1 (2 bits), S,2 (4 bits), and U,3 (5 bits) (Fig. 4D and Supplementary Fig. 9).

## The 3D wireframe DNA origami encryption, decryption, and fast readout

Implementing DNA origami cryptography in 3D nanostructures enhances security by firstly expanding the unused folding paths key space if only 2D origami is being used to encrypt the information, and secondly concealing the encrypted pattern within complex 3D architectures, as opposed to the prior approach using 2D origami nanostructures[31]. Several groups have demonstrated various 3D DNA origami shapes, including the cuboctahedron (Figs. 1B and 5), which has a radius of ~35 nm[33,54–57]. Revealing the true shape of and the encrypted pattern on 3D DNA origami required 3D super-resolution imaging with sub-25 nm resolution. In comparison, AFM, which is restricted to 2D imaging, may misinterpret the 3D wireframe cuboctahedron as a 2D construct, leading to potential confusion about the encrypted pattern (Fig. 1B and Supplementary Fig. 10). We identified the high-speed DNA-PAINT, capable of imaging 3D patterns on 3D DNA origami with ≤10 nm $z$-resolution[44,58,59], as a superior readout technique for 3D DNA origami cryptography.

To demonstrate DNA origami cryptography in 3D, we encrypted the date 0407 (4th of July) on a 3D wireframe DNA origami cuboctahedron template. In our design, the docking/information strand binding length to the scaffold ranges from 52–54-nt (Supplementary Table 5). The pattern encryption allocated 4 bits (ABCD) for number encryption, 3 bits (EFG) for position encryption, allowing a maximum of 8 characters being encrypted, and 5 bits for alignment markers (3 filled and 2 empty) to break the in-plane ($xy$) rotational symmetry (Fig. 5A). The bits are located at the vertices of the cuboctahedron,

global bit alignment maps for each letter (Fig. 4B) demonstrated that with one redundancy, the main bits do not need to be 100% present to retrieve the correct letter information, as the redundant bits support the main bits. In the correct readout map, the main bits are not always 100% present, but the redundant bits compensate for this. The incorrect readout map shows that some dockings without information strands have been falsely assigned a 1 bit. The combined readout of the

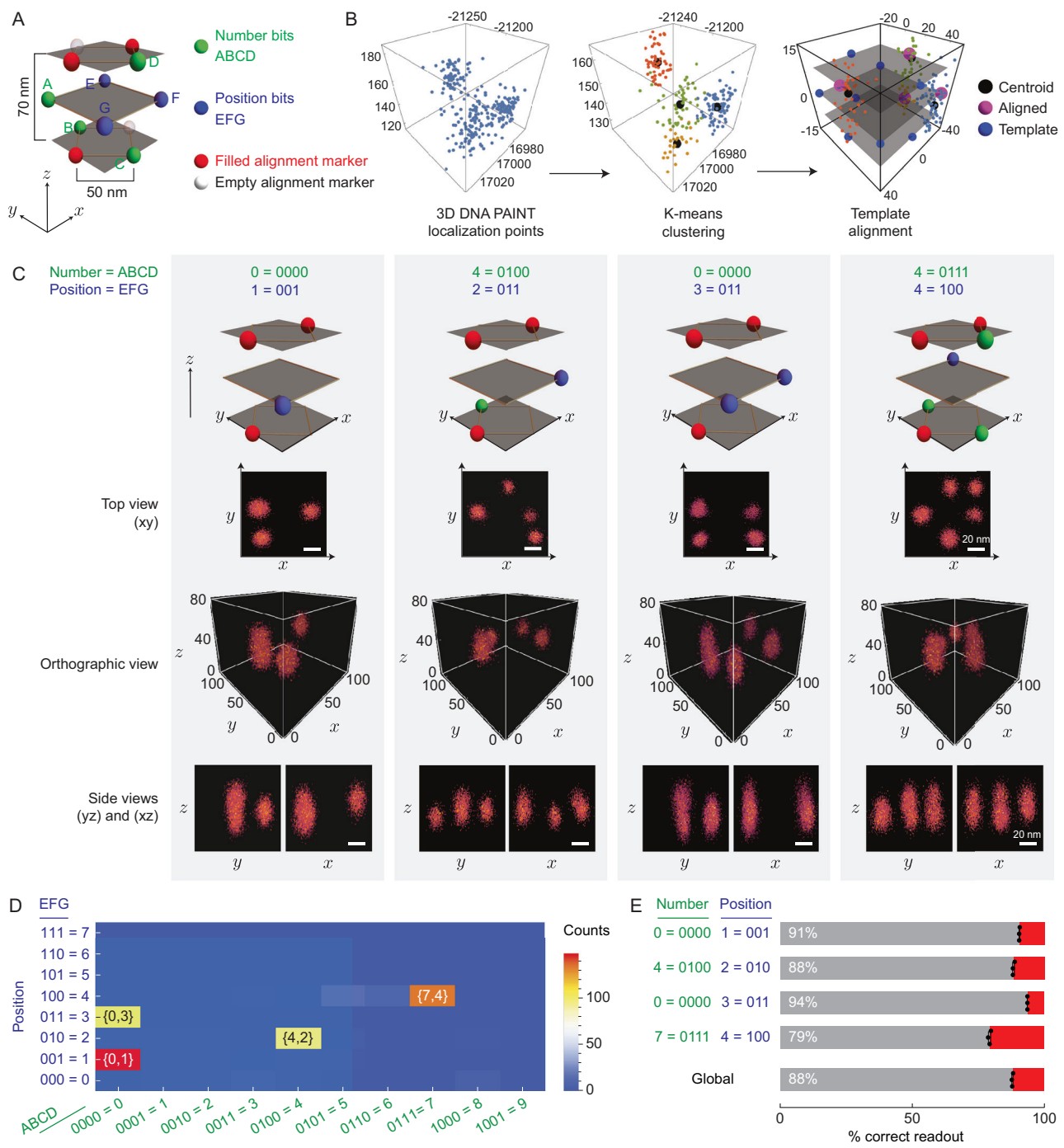

**Fig. 5 | 3D DNA-PAINT encryption applied to information encoded in a 3D wireframe DNA origami cuboctahedron. A** Encryption rule for the pattern comprising number bits (green), position bits (blue), and alignment markers (red and white). Three z planes are colored transparent gray, occupied by the corner points. **B** Example of 3D DNA-PAINT localization data from a pattern on the 3D wireframe cuboctahedron DNA origami (left), k-means clustering result (middle), and 3D alignment with a cuboctahedron template (right). **C** Superposition images of

correctly aligned and read 4 patterns from identified 3D DNA origami ($N = 161, 107, 108$, and $168$, left to right, respectively). Scale bars: 20 nm. **D** One run of readout presented as positions vs. numbers vs. counts, showing 4 main peaks of letters. **E** Individual letter readout mean percentage of correct (gray) and wrong (red) readouts with uncertainties (mean values ± SD with $N = 3$ runs of the same dataset processed as in (**B**)).

with the possibility of placing more bits on the edges. The distance between 2 stacking bits along the z axis is 65–70 nm based on the ideal design (Supplementary Fig. 11). We acquired 43,510 frames with an exposure time of 50 ms, resulting in a total acquisition time of ~35 min. We then selected 150–210 3D DNA origami structures for each pattern using visual inspection (Supplementary Fig. 12).

Analysis of the localization data for each selected DNA origami (Fig. 5B) revealed the challenges in resolving the vertical (z) separation of 60–70 nm with our experimental setup. Distinguishing 2 clusters in the localization data is difficult, as localizations from a docking spot occupy a 3D space with a z value range of approximately 50–60 nm (Fig. 5B, C). The flexibility of 3D wireframe origami, as revealed by AFM,

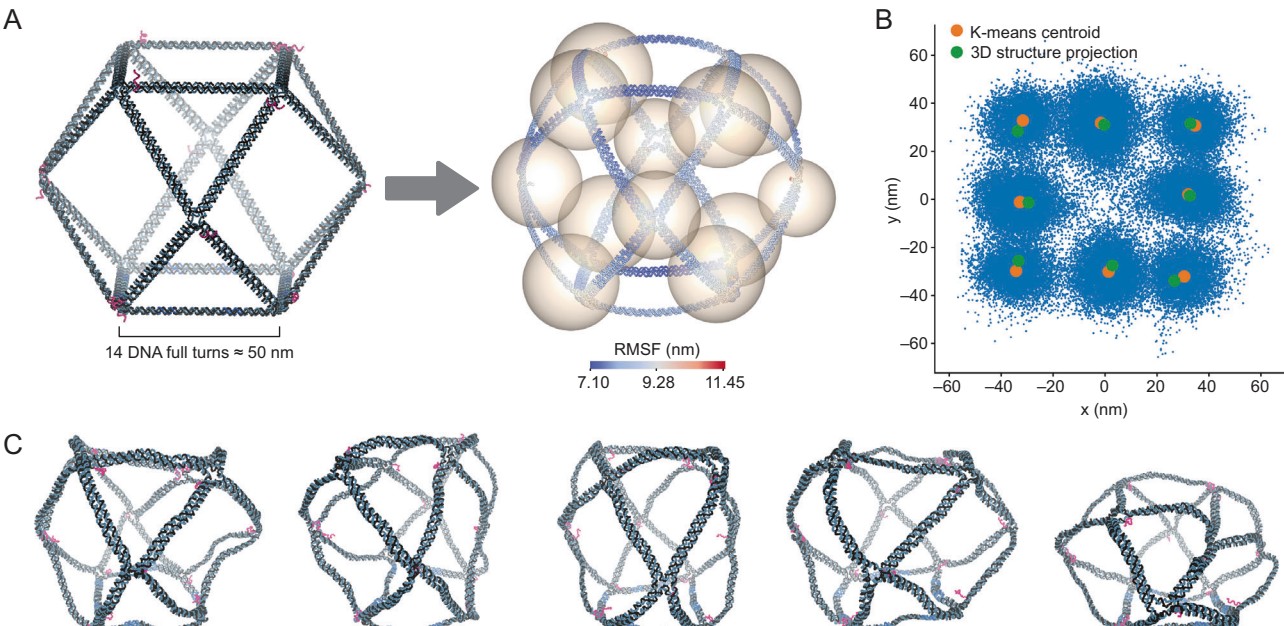

**Fig. 6 | 3D wireframe cuboctahedron oxDNA and DNA-PAINT alignment with the mean structure. A** Orthographic views of the original unrelaxed 3D cubocta-hedron structure with red-colored docking handles (left) and the mean structure after oxDNA simulation, with nucleotide coloring based on RMSF (right). Spheres represent DNA-PAINT data, with centers based on centroids of clusters after 3D k-means clustering and radii based on the average distance of each localization to its centroids. Spheres fitting with the mean structure are obtained after scaling the $z$ value of DNA-PAINT data by 3.3. **B** 2D projection of DNA-PAINT localizations from 3D cuboctahedrons ($N = 173$). The k-means centroids (averaged if there are 2 stacking centroids along the $z$-axis) are shown by orange disks, while the 2D projections of the docking ssDNA centers post-oxDNA simulation (similarly aver-aged if there are 2 stacking projections along the $z$-axis) are denoted by green disks. **C** Time lapse from the oxDNA trajectory of the cuboctahedron reveals structural fluctuations. The dockings are colored in red, and biotinylated strands are colored in light blue.

further compromised the $z$ resolution. Comparing our results to pre-vious studies is challenging, as no groups have attempted 3D DNA-PAINT on single-scaffolded 3D wireframe DNA origami, and the $z$ separation of <70 nm is beyond the state-of-the-art resolution achieved by many groups[58–60]. Despite the imperfect 3D DNA-PAINT localization data, the k-means clustering and elbow method could still assign centroids (Fig. 5B(middle)). However, the ~30 nm separation between the 2 centroids from the k-means clustering did not conform to the designed docking position, which we attribute to the large spread of the $z$ value of localization data for 2 stacking clusters com-pared to the $x$ and $y$ spread of 2 neighboring clusters.

The averaged 3D localization data images post-alignment indi-cated that the top and side perspectives of each pattern exhibited a high degree of match with the schematic in the top row of Fig. 5C. The side views demonstrated a clear height difference between a single cluster and 2 stacking clusters. Despite the challenge of obtaining clearly resolved 2 stacking dockings, we successfully aligned the cen-troids with the template (Fig. 5B). The combined readout shows 4 distinct peaks corresponding to the encrypted numbers 0,1, 4,2, 0,3, and 7,4, with minimal false readout peaks in the background (Fig. 5D). By employing the method in Fig. 5B with the addition of alignment marker filtering, we extracted the bits and retrieved the encrypted information with a global accuracy of 88% (Fig. 5E). The alignment marker filtering was a necessary step due to the lack of redundancy in the alignment markers compared to the 2D origami pattern encryption rules. Our result demonstrated, for the first time, the successful application of 3D DNA-PAINT on a 3D wireframe cuboctahedron DNA origami as a method for 3D DNA origami encryption readout, achiev-ing complete information retrieval.

Coarse-grained dynamic simulations using oxDNA[46,47,61,62] indi-cated the flexibility inherent in the 3D DNA origami cuboctahedron, with the mean structure having root mean squared fluctuations (RMSF) ranging from 7.1 to 11.45 nm. The vertices were more rounded

and the edges bent, accounting for the observed compromise in the resolution along the $x$, $y$, and $z$ axes in the 3D DNA-PAINT data (Figs. 5C and 6A). Aligning the 3D DNA-PAINT localization data ($N = 173$ parti-cles) bearing all vertices' docking with the docking center points in the mean structure, using 3D k-means clustering (Supplementary Fig. 13A) and additional $z$-axis scaling, resulted in an alignment with a root mean squared distance (RMSD) of 100.88 nm (Fig. 6A(right)). However, alignment on the 2D projection of the 3D DNA-PAINT data and the mean structure produced better results, characterized by an RMSD of 26.07 nm (Fig. 6B and Supplementary Fig. 13D), indicating that the oxDNA simulation agreed well with the experimental DNA-PAINT data in the $xy$ plane without the need for scaling and the discrepancy was predominantly confined to the $z$ dimension. The intrinsic flexibility of the 3D wireframe DNA origami cuboctahedron becomes increasingly evident upon examination of the oxDNA trajectory (Fig. 6C). Com-paring the 3D alignment of DNA-PAINT in the mean structure with the alignment in the unrelaxed structure (Supplementary Fig. 13B, C, and E) corroborated the interpretation that the structure exhibits sig-nificant flexibility and agrees better with the mean structure obtained from the oxDNA simulations.

We further performed control experiments using a more rigid 3D DNA origami tunnel structure (Supplementary Fig. 14 and Supple-mentary Table 6)[58]. The results from the DNA-PAINT experiments with DNA origami tunnels showed that the measured distances between dockings along the $z$-axis (~73 nm), closely matched the expected 75 nm separation as predicted by oxDNA simulations (Sup-plementary Fig. 14). These findings indicate that the observed dis-crepancy in the dimensions of the 3D DNA-PAINT images of DNA origami cuboctahedrons along the $z$ direction was due to structural flexibility, rather than to any limitations inherent in the 3D DNA-PAINT imaging process itself. These results demonstrated that rigid 3D structures are essential for achieving accurate spatial readout in DNA origami cryptography.

## Encrypting information in a multimer structure with a rigid 3D DNA origami tetrapod dimer

To address the potential resolution limitations caused by structural flexibility, we demonstrated the encryption of information using a rigid 3D DNA origami structure known as a tetrapod (Fig. 7 and Supplementary Table 7)[63]. The rigidity of the tetrapods arises from their 24-helix thick arms, in contrast to the 2-helix arm of the DNA origami cuboctahedron in Fig. 5. To assess rigidity of tetrapods, we placed 4 docking strands at the end of each tetrapod arm (Fig. 7A) and performed 3D DNA-PAINT imaging. Three-dimensional DNA-PAINT images showed that the measured distances between the 4 docking spots in both the $xy$ plane and the $z$ direction match the design (Fig. 7B). To encrypt information, we performed dimerization of the tetrapod through one of the arms by connecting the sticky ends (Fig. 7D) and constructed the encryption rule for the pattern, comprising number bits, position bits, and alignment markers (Fig. 7E). Dimerization was required to increase the information capacity beyond 4 bits, thereby enabling the encryption shown in Fig. 7E. We encrypted the information 1776 (Fig. 7G(top), and Supplementary Table 8 for the mixing of strands to form the monomer corresponding to the pattern), so that combining cuboctahedron and dimer tetrapod encryption encodes the information for July 4th, 1776. As a proof of concept, we performed the experiment by individually synthesizing and purifying 2 monomers corresponding to each specific pattern prior to dimerization of the DNA-PAINT sample. We selected the best single molecule for each dimer pattern from the DNA-PAINT experiment, where the experimental results match the designed pattern (Fig. 7G(bottom), and Supplementary Fig. 15), showing the rest of the selections). Dimerization added another layer of security to form the correct pattern in which information is encrypted; a correct dimer must form. These results validated that polymerization can be used to increase the number of bits, allowing more characters to be encrypted and adding an extra security layer[31].

## Discussion

In this study, we demonstrate a DNA origami cryptography protocol that significantly increases the practicality and potential for secure information storage and transmission using 2D and 3D DNA origami structures. Our protocol leverages 3D DNA origami structures and fast DNA-PAINT readout to achieve high-density and secure data storage for communication. Using high-speed DNA-PAINT, we achieved a >6-fold increase in data acquisition time compared to the previously published DNA-PAINT readout of DNA origami information storage[21]. Building on the foundational work by Dickinson et al.[21] and Zhang et al.[31], our approach offers several key advantages, including increased information density, improved design criteria, faster imaging and readout speeds, enhanced geometric complexity and up to 89% accuracy in data retrieval. By exploiting the vast possibilities of scaffold routing, we generated a large key space for secure information encryption and retrieval in both 2D and 3D nanostructures. Furthermore, we demonstrate the crucial role of error-correction codes in maximizing the chance of complete message retrieval, enabling the encryption of information using a one-redundancy code with 48 dockings per origami.

Using the results obtained in our study, design criteria that have never been discussed before in the DNA origami cryptography proof of concept work by Zhang et al.[31] can be applied to achieve high information retention and retrieval with high-speed pattern readout using the DNA-PAINT technique combined with unsupervised clustering and template alignment. We recommend having a docking strand binder to be at least 32 nt, with the longer, the better (Fig. 2) for high efficiency of information retrieval through DNA-PAINT and the readout protocols. The assignment of dockings is also important. We suggest utilizing the dockings in the center area of a 2D structure for higher incorporation and detection efficiency, thus giving higher information retention and

retrieval (Fig. 2B–E). For 3D origami, we recommend a minimum $z$ dimension of 100 nm, given the axial resolution limit of 3D DNA-PAINT. The redundancy of bits is significant when dealing with high-density docking origami (Fig. 4). We recommend having an error-correction code to maximize the chance of obtaining the complete message. In our case of a one-redundancy code with 48 dockings per origami, by using 12 dockings at corners for alignment markers, 36 dockings are available for the information bits, which are then split into 6 bits for letters, plus 6 bits for the redundancy, and 12 bits for positions, plus 12 bits for position bits redundancy. The 12 bits correspond to a total of 4096 letters, which is equivalent to 2 pages of text information (assuming that 1 page can have 2000 characters). The possibility of 2D and 3D geometries will increase the complexity of scaffold routing and the possibility of staple strands interlacing; thus, we expect to have key sizes larger than the estimated 700 bits for the commonly used M13mp18 scaffold from Zhang et al.[31]. Our protocol also improves information density, design criteria, imaging and readout speed, and especially geometric complexity for the benefit of security. Since the key size depends on the length of the scaffold, another possible avenue to increase the key size is to use several orthogonal scaffolds with custom-made sequences, thus enabling larger DNA origami. Multi-orthogonal scaffolding of larger DNA origami has been demonstrated[64]. As an implication, with larger DNA origami and the same information density per origami, more information strands can be incorporated, thus giving a greater total information that can be encrypted in a specific DNA origami of multiple orthogonal scaffolds. Using unique custom orthogonal sequences can further enhance security, as these sequences are different from commercially available scaffolds, which are derived from the M13mp18 scaffold.

Although our protocol exhibits high accuracy in information retrieval, there is still room for improvement in resolution and density. First, state-of-the-art DNA-PAINT has achieved Ångström[44] and sub-5 nm[45] resolution, surpassing the results demonstrated in this work. To achieve this resolution, shorter binding lengths between the docking strands and the scaffold would be required, potentially compromising the retention of the docking/information strands on the origami. To mitigate this limitation, we propose the use of modified information strands or scaffolds with covalent linkers[64,65], thereby strengthening the binding affinity and ensuring robust information retrieval. Second, to further increase information density, multicolor DNA-PAINT can be used[45,58]. For instance, in the case of a 2D RRO template with 48 docking strands per origami, using 2 fluorophores such as ATTO 488 and ATTO 647 for 2 orthogonal high-speed docking sequences would allow for an additional 36 docking spots (assuming that 12 docking spots of the new color are also being dedicated for alignment markers). These docking spots can be used for 18 additional position bits and 18 redundancy bits, collectively resulting in 28 position bits, capable of encoding ~268 million characters (Supplementary Fig. 16). It should be noted that the orthogonality of the 2 imager strands labeled with different fluorophores has to be met to prevent spurious or cross-interactions between the imagers. The spurious interaction between imagers can negatively impact DNA-PAINT super-resolution imaging, manifested in the increased background, thus compromising resolution.

A practical limitation in terms of readout has also to be noted. While the proposed origami design presented in Supplementary Fig. 14 can accommodate millions of characters to be encoded, performing a readout of millions of characters is practically challenging in terms of the readout time. The challenge is to have enough origami to be imaged so that a reasonable number of origami for each pattern can be collected for decoding or decryption. With a global accuracy of around 75–89%, we would want to collect at least 100 origami images for each pattern. This means that for one million letters, we would need to acquire 100 million origami images. High-speed DNA-PAINT is a high-throughput method that can capture approximately 5000–10,000

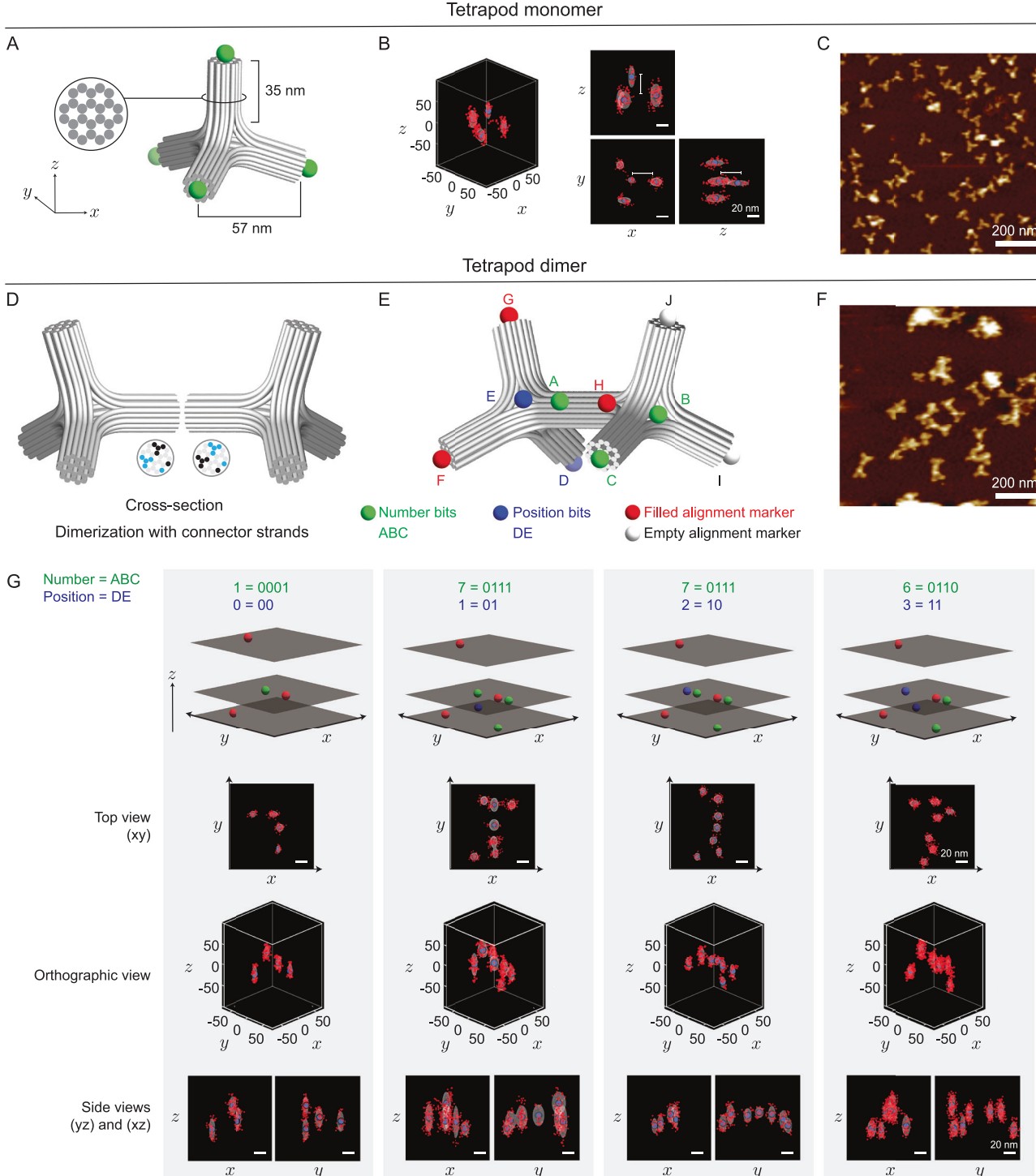

**Fig. 7 | 3D DNA-PAINT encryption using rigid tetrapod DNA origami dimers. A** A tetrapod DNA origami monomer showing docking sites (green spheres) on one face with each arm. (Inset) Each arm is rigid and comprises 24 DNA helices.
**B** Representative 3D DNA-PAINT localization of a tetrapod, shown in orthographic and 2D projection views. Scale bars are 20 nm and thin white lines are 35 nm.
**C** Representative AFM image of tetrapod monomers. **D** Dimerization of 2 tetrapod monomers via connector strands, with cross-sections highlighting the joining interfaces. The binding sites are shown as blue and black circles in the cross-section.

**E** Encryption design of a tetrapod dimer with number (green), position (blue), and alignment (red and white) bits. **F** Representative AFM image of tetrapod dimers. **G** Representative single-molecule images of DNA-PAINT readouts for 4 dimer patterns, in agreement with the designed patterns. The gray colored voxel volume is generated by summing 3D Gaussian functions centered at each cluster centroid, with widths determined by the spatial standard deviation of points in each cluster. The blue spheres represent the centroids of each cluster obtained via k-means clustering. Similar results are obtained in 3 experiment repeats.

origami per field of view (500 × 700 μm$^2$) in a single imaging session per microscope. Therefore, obtaining 100 million origami molecule images would require 10,000 imaging sessions, which is practically challenging in terms of the readout time. We would say the current practical limit is to perform 10–20 imaging sessions where we can obtain a maximum 200,000 origami that can accommodate around 2,000 letters, and these imaging sessions would consume around 6–12 h if a session takes 35 min, as we described in the manuscript. To scale it up, additional microscopes can be employed to expedite the process. Furthermore, we compared the nanoscale readout modalities of AFM, DNA-PAINT, and nanopore for DNA nanostructure-based data storage where DNA-PAINT excels in dimensional capability and resolution with comparable throughput to nanopore (Supplementary Table 11).

Another notable demonstration is the improvement in data density per origami. Our current experimental realization has the same docking density of 48 dockings per origami as in the work by Dickinson et al.[21], which translates to a maximum of 2 bytes per origami data density. This density is already sufficient to encrypt many data types, such as char, bool, unsigned int16, int16, unsigned int8, int8, signed char, unsigned char, short, unsigned short, wchar t, all of which occupy only 1 or 2 bytes for one value[66]. However, increasing the data density per origami is also useful in the case of encryption of several data types, such as int or float, which occupy 4 bytes for a value. To increase the data density per origami, in addition to the two-fluorophore strategy mentioned above with slight modification of the encryption pattern layout to achieve 4 bytes per origami data density, making the separation between docking spots smaller than 5 nm also results in more docking spots per origami, thus achieving a higher data density per origami of 4 bytes per origami. We showed the simulated DNA-PAINT using the Picasso Simulate module[67] to demonstrate the possibility of a data density of 4 bytes per origami (Supplementary Fig. 17).

In the case of 3D DNA origami, since DNA-PAINT requires TIRF, which employs evanescent waves to obtain high-resolution reconstructions, the limit of the 3D origami height that can be imaged follows the decay length of the evanescent wave, which is around 150-200 nm from the surface. In an ideal case with the state-of-the-art DNA-PAINT that can achieve around 1 nm resolution in $x$, $y$, and $z$ resolution[44], we can estimate that with an ideally rigid 3D structure of a tunnel shape that can be formed with a single scaffold with a typical length of 8000 nt. The origami will have a height of 75 nm and side lengths of 16 nm, and we can design a 5 nm separation in $z$ direction and 2.5 nm in $x$ and $y$ directions for the information strands. Therefore, we can accommodate around 15 bits along the $z$ direction with a height of 75 nm and 20 bits along the $x$ and $y$ directions. In total, we can accommodate $15 \times 20$ bits, which is equal to 300 bits for a single origami formed from a single scaffold. The 300 bits can then be distributed for a specific number of alignment markers, bits for characters, and position encryptions. A particular example case for the distribution of these 300 bits is 10 bits for alignment markers, 8 bits for the ASCII, and adding 2 redundancies totaling 24 bits, and the remaining 266 bits can be used for position encryption, which is enough to encrypt $2^{266}$ characters. This particular simple analysis with ideal assumptions was performed to show the physical limitations on the total number of characters that can be encrypted, assuming that we can achieve ideal DNA-PAINT super-resolution imaging and structural rigidity conditions using specific 3D DNA origami. Longer scaffolds can be exploited to create a larger origami in the $x$, $y$, or $z$ directions, thus providing more bits along those directions.

We additionally conducted an experiment to highlight the necessity of a complete set of staple strands for proper folding. The nanostructure was assembled by combining the M13mp18 scaffold with docking sequences for the 10 nm grid (key 2), while omitting the remaining staples (key 3). DNA-PAINT imaging and subsequent super-resolution reconstruction revealed no 10-nm grid patterns (Supplementary Fig. 18). This confirms that using only key 2 with 48 information strands and 64-nt binders, totaling 3072 bp of hybridization with the 7249-nt scaffold, was insufficient to generate the intended pattern. Thus, without the full complement of the staple strands, the correct structure and encoded pattern do not emerge. This finding underscores that the remaining staples must follow a highly specific set for successful folding. Consequently, attempting brute-force approaches to predict all the possible folding paths is practically infeasible.

For data classification, we conducted a comprehensive evaluation of multiple methodologies, ultimately adopting the k-means clustering proposed by Huijben et al.[68], which was predicated on its efficacy in 2D and 3D space, coupled with its direct implementation and fast computational performance. Alternative methodologies, including DBSCAN, mean-shift clustering, and Gaussian mixture models, require the fine-tuning of numerous parameters tailored to the dataset. Conversely, the k-means approach primarily hinges on determining the optimal number of clusters by applying the elbow method. The method employed by Huijben et al. has a runtime that increases quadratically with the number of origami owing to the requirement of pairwise registration to create the dissimilarity matrix, whereas our method has a linear runtime. We acknowledge that using their classification methodology enabled us to achieve an enhanced accuracy of ~90% for the NSF dataset, albeit at the expense of speed. However, we encountered limitations in accurately decoding the ASU dataset with one redundancy (Supplementary Fig. 19). Furthermore, we explored supervised learning classification frameworks, such as MLP[48] and ResNet convolutional neural networks (CNN)[69], as demonstrated in (Supplementary Fig. 20). These models require a training dataset that encompasses all possible permutations of the pattern for each encryption rule, rendering the approach impractical. Specifically, for our ASU pattern encryption rules, which involve 24 information bits, the generation of approximately $2^{24}$, or roughly 16.7 million, patterns would be required for the training dataset. Our readout method, which employs k-means clustering alongside template alignment via minimization, eliminates the need for training data and facilitates a fully automated readout process when integrated with auto-picking. Further automation and speed improvement are feasible through the application of DNA-PAINT augmented by deep neural networks[70].

In conclusion, our DNA origami cryptography protocol significantly advances the field of molecular and DNA cryptography by offering new design considerations and demonstrating the versatility of our approach by exploring alternative machine learning methods for readout. The successful integration of 3D DNA origami structures, fast DNA-PAINT readout, and machine learning algorithms for data classification demonstrates the power of interdisciplinary collaboration in advancing the frontiers of biophysics and computer science. Our work shows the potential of secure and high-density information storage and communication using DNA nanostructures, representing a crucial step forward in developing practical and secure methods for data storage and transmission at the molecular scale. As we continue to explore the vast potential of DNA as a medium for information storage and transmission, future research could focus on further increasing the information density and security of the protocol by integrating advanced techniques, such as multicolor DNA-PAINT, the utilization of information strands or scaffolds composed of modified DNA with covalent linkers, and the application of deep neural networks for faster automation. The implications of our work extend beyond cryptography, with potential applications in various fields including healthcare, data storage, and computing, highlighting the interdisciplinary nature of this research and its potential to inspire new developments in biophysics, computer science, and applied mathematics. Ultimately, our study lays the foundation for future research and development in the rapidly evolving field of DNA-based information storage.

## Methods

### Materials

Unmodified DNA staple strands and biotinylated staple strands for 2D and 3D origami were obtained from Integrated DNA Technology (IDT). M13mp18 and P8064 scaffold strands were ordered from Bayou Bio-labs, while p8634 scaffold was purchased from Tilibit Nanosystems. Amine-modified DNA strands for imager strands were purchased from IDT DNA. Cy3B-NHS Ester fluorophores (PA63101) were ordered from General Electric Healthcare. Chemicals were purchased from Sigma-Aldrich. Glass coverslips (48466-205) and microscope slides (16004-430) were purchased from VWR. Kapton tape (PPTDE-2) for the flow chamber was purchased from Bertech.

### Imager strands

Imager strands were prepared by conjugating amine-modified DNA strands to Cy3B using NHS Ester coupling. The conjugated imager strands were then purified using high-performance liquid chromatography (HPLC).

### 2D and 3D DNA origami folding and purification and tetrapod dimer assembly

We used the M13mp18 scaffold strand to fold 2D Rothemund rectangle and 3D cuboctahedron, P8064 scaffold strand to fold 3D tunnel, and P8634 scaffold strand to fold tetrapod DNA origami. In general, for the specific structure, the corresponding scaffold was mixed with staple, biotinylated strands, and the corresponding docking strands for each pattern in 1X TAE 12.5 mM $Mg^{2+}$ buffer. M13mp18 and P8064 scaffold strand concentrations were 20 nM while P8634 concentration was 30 nM. The staple strands were at a 10X concentration compared to the scaffold. To optimize the incorporation of docking and biotinylated strands, an excess concentration of 30–60X was used[52]. 3D tetrapod origami was annealed with 20 mM $MgCl_2$, while other structres were annealed at 12.5 mM $MgCl_2$. The mixture was annealed by ramping up to 80 °C for 5 min, followed by a slow ramp down to 4 °C at a rate of 3 min 12 s per degree Celsius, based on the protocol by Schnitzbauer et al.[67]. After annealing, the DNA origami solution was purified using a PEG precipitation method with a PEG buffer (15% (m/v) PEG 8000, 12.5 mM $MgCl_2$ (20 mM $MgCl_2$ for tetrapod origami), 505 mM NaCl, 5 mM Tris-HCl, 1 mM EDTA, pH = 8.0) for three rounds[71]. Finally, the solution was redispersed in 1X TAE 12.5 mM $Mg^{2+}$ (20 mM $Mg^{2+}$ for tetrapod origami) buffer and stored at −20 °C. The entire process encompasses the decryption of DNA origami through origami folding.

The dimer growth solution was prepared in a PCR tube by mixing both tetrapod monomers in $1 \times$ TAE buffer containing 20 mM MgCl. Tetrapod monomers were designed with complimentary sticky ends extending outwards to improve specificity and selectivity so that when two tetrapods align face-to-face hybridization took place simultaneously. A face of each monomer contained 2 pairs of complimentary sticky ends (staples and mixing details in Supplementary Tables 7 and 8). Dimerization was initiated by incubating the dimer mixture at 37 °C and cooled down to 25 °C at a rate of 10 min and 12 s per degree Celsius. A 30 min hold at 25 °C was performed to stabilize dimer binding. The assembled dimers were used for DNA-PAINT imaging without purification.

### AFM imaging of 2D and 3D DNA origami

Bruker multimode AFM was used to image DNA origami after PEG purification. 3 μL of 1–2 nM DNA origami solution was deposited on a freshly cleaved mica surface on an AFM metal sample slab. Then, 60 μL of 1X TAE 12.5 mM $MgCl_2$ (20 mM $MgCl_2$ for tetrapod origami) and 4 μL of 0.2 M $NiCl_2$ solution were added to immobilize the origami onto the mica surface. This sample was imaged in the AFM using fluid mode with SCANASYST-FLUID+ model tip by Bruker. The image was processed using NanoScope Analysis AFM software.

### Pattern readout by 2D and 3D DNA-PAINT super-resolution imaging and image processing

DNA origamis with specific patterns such as N, S, F (Fig. 3), A, S, U (Fig. 4), 0, 4, 0, 7 (Fig. 5) and 1, 7, 7, 6 (Fig. 7) were immobilized on a different BSA-biotin-streptavidin coated coverslip that forms a flow chamber with microscope slide glass through double-sided Kapton tape. Buffer A+ (10 mM Tris-HCl, 100 mM NaCl, 0.05% (v/v) Tween 20, pH 8.0) was used to dilute BSA-Biotin and streptavidin to 1 mg/ml and 0.5 mg/ml concentration, respectively. Buffer B+ (5 mM Tris-HCl, 10 mM $MgCl_2$, 1 mM EDTA, 0.05% (v/v) Tween 20, pH 8.0) was used to dilute DNA origami at experimental concentrations. Buffer B+ was also used to dilute the imager strands (AGGAGGA/3' Cy3B/) to experimental concentrations. Oxygen scavenger solutions PCA, PCD and Trolox with final concentrations of 1.25X PCA, 1X PCD, and 1X Trolox were mixed with the imager strands to make the final imaging solution. Individual letter or pattern was imaged separately so that the accuracy study can be carried out by knowing that all the picked origamis from specific letter sample is coming from single pattern thus there were no false negatives and false positives which can interfere the data retrieval accuracy results. In the real scenario, mixing all patterns can be done to simplify the imaging steps. The experimental conditions for each figure are described in the Supplementary Table 5.

We used an Oxford nanoimager (ONI) of a Benchtop Nanoimager S Mark II with a total internal reflection fluorescence (TIRF) setup. Cy3B was excited using a 532 nm laser with a power density ranging from 800 to 1250 W/cm². A 549–623 nm Bandpass filter was installed on the emission path to select Cy3B emission. For the NSF dataset, an Olympus objective with 100X magnification and 1.4 NA with oil immersion was used. For the ASU, 0407, and 1776 dataset, an Olympus objective with 100X magnification and 1.49 NA with immersion oil was used. A Hamamatsu ORCA-Flash4.0 V3 digital sCMOS camera was used to acquire the DNA-PAINT movies with a camera exposure time of 50 ms. The microscope was equipped with z-lock autofocus with a piezo stage. For 3D DNA-PAINT, an additional 3D lens was inserted into the optical path that modifies the point spread function (PSF) from circular to elliptical, which is dependent on the z distance of the emitter from the focal spot through astigmatism[59,60,67,72]. The z-calibration was done by using 2D RRO by scanning the z from −500 nm to +500 nm with an increment of 10 nm made possible by the piezo stage. A standard 3D calibration curve (Supplementary Fig. 21) was generated through the Picasso Localize module, which was used as a calibration curve to process the 3D DNA-PAINT movies. The imaging conditions for each figure are described in Supplementary Table 5.

The DNA-PAINT movies were processed using FIJI[73] to crop the image into 256 px by 256 px. Then, each movie was fed into the Picasso Localize module to fit the PSF into precise localization data (Supplementary Table 9 for Picasso Localize module parameters). The Picasso Render module was used to perform multiple redundant cross-correlation (RCC) drift corrections to obtain the final super-resolution images. The Picasso Filter module was used to remove outlier localizations based on x and y localization precision and localization background. The whole processing was performed on an Alienware Desktop Computer with an Intel Core i7-6800K CPU, 32 GB RAM, and NVIDIA GeForce GTX 1080 graphics card.

### 2D DNA-PAINT incorporation efficiency analysis

Picasso processed localization data of super-resolution images of the patterns using the Picasso pick automatic function. All picks were aligned using the Picasso Average module. The aligned picks were then unfolded in the Picasso Render module to get arrays of picks that had been aligned in the same orientation. Then, it was fed into a Matlab script that analyzes the incorporation efficiency (Supplementary Fig. 3).

## 2D and 3D data clustering, template alignment, and analysis

The protocol followed the pipeline as shown in Fig. 3A. The patterns on each origami were picked by visual inspection on the Picasso Render module. The picks were then used as input for the following processing of K-means clustering, which assigns centroids based on the optimal number cluster followed by cluster filtering. Based on the pattern encryption rules, the centroids were aligned with their corresponding templates. The processing was done in a System 76 Thelio Desktop with AMD Ryzen 9 5950X 16-Core Processor, 32GB RAM, and NVIDIA GeForce RTX 3070 GPU. The detailed explanations for each process are as follows.

### K-means clustering

The clustering was performed using the Scipy Python package[74] for the 2D data and a Matlab function for the 3D data. In the 3D data, extra data filtering was performed using the DBSCAN function in Matlab by empirically assigning the radius ($\epsilon$) to be 7 and the minimum number of neighbor points to determine the core to be 4. In the implementation, k-means clustering was used to cluster localizations into $N$ through $M$ clusters, where $N$ and $M$ were chosen according to the pattern encryption template. K-means clustering takes one parameter k for the number of clusters. The optimal k value was determined using the elbow method by running k-means for ($M - N$) times over each origami, initially with $N$ clusters and increasing by 1 every iteration until $M$ clusters, where M was determined by the maximum docking numbers for specific templates. The inertias, centroids, and cluster memberships resulting from the k-means clustering were recorded at each iteration. Centroids are defined as the center of each cluster found by k-means. Inertias are defined as the squared distances between each localization and its corresponding centroid. The differences between the inertias, at each iteration, were termed gradients, and they were calculated, then negated and normalized to have a maximum value of 1. The optimal number of clusters was chosen by comparing these differences and locating the point at which the difference negligibly improves. The optimal number of clusters was the point at which the gradient value starts to saturate to a value of 0.95 or more (maximum value is 1) in the case of 3D data or closest to some threshold $T_I$ in the case of 2D data. See Supplementary Table 10 for the parameters. The elbow method is a relatively standard practice.

### Cluster filtering

The filtering was done after k-means clustering over the 2D data. Outlying clusters were filtered out on the basis of size, defined as the number of localizations belonging to that cluster. To do this, the mean size of all clusters was calculated and the size threshold was calculated by multiplying the mean size by some value $T_S$. Clusters smaller than the threshold size were discarded and localizations belonging to those clusters were filtered out.

### Template creation

The process was performed based on the pattern encryption rules of 2D and 3D DNA origami. Four templates were created for the grids used in the NSF dataset, the ASU one redundancy dataset, the ASU two redundancy dataset, and the 3D cuboctahedron 0407 dataset. These templates use prior knowledge of the possible locations of the binding site and orientation marker positions. Each point in the template represents the approximate position ($x$, $y$) of the binding site in an idealized origami. Note that all binding sites are included, not just those used in any particular pattern.

### 2D template alignment

The centroids, template, cluster sizes, template weights, and orientation markers were used to calculate a transform of the template that minimizes the following cost function modified from the Euclidean squared distance formula:

$$C = \frac{\sqrt{D+1}}{n} \sum_{i=1}^{n} (|A_i - B_i| P_i S_i)^2, \tag{1}$$

where $A_i$ and $B_i$ are the Euclidean coordinates of the $i$th centroid and the closest point in the transformed template to $A_i$ by Euclidean distance, respectively. $S_i$ is the number of localizations in the ith centroid's cluster. $|A_i - B_i|$ is the Euclidean distance between $A_i$ and $B_i$. $P_i$ is the weighting of $B_i$ defined as $P_i = W_0$ if $B_i$ is an orientation marker, otherwise $P_i = 1$. Finally, $D$ is the number of orientation markers that are not the closest point to any centroid.

We assume there is a higher probability that binding sites lie closer to larger clusters; hence, the cost penalty is larger if the template is misaligned from larger clusters. In addition, the orientation markers are very important for readout, so these points in the template are weighted at $W_0 > 1$, and any missing orientation markers are penalized with the $D + 1$ term. Besides these additions, the cost function is a squared nearest-neighbor distance metric.

The transform parameters include rotation by an angle $\theta$, $X$ or horizontal scaling, $Y$ or vertical scaling, $X$ translation, and $Y$ translation, which are applied in that order. The rotation prior to scaling allows the transformed templates to shear in response to distorted origami. At each iteration of optimization, the current transform is applied to the template, then the cost is calculated.

$C$ is minimized according to any optimization method, and the final transform is applied to the template. For the NSF dataset, the template was first aligned to the origami by translating and rotating the template at fixed intervals around the origami until the minimal distance was achieved. The rough transform was then fine-tuned using the L-BFGS-B method of the minimize function from Scipy[74]. For the ASU 1-redundancy and 2-redundancy datasets, the transform was directly optimized using the differential evolution method of the minimize function from Scipy. For both datasets, at each iteration of the Scipy algorithm, all transformation parameters were simultaneously optimized and applied in the order specified above. Parameter selection was done by empirically finding good values for $N$ and $M$. If origami with a larger number of binding sites is imaged, a higher value of $M$ may be needed to account for false positives. $T_I$ and $T_S$ were selected through grid search. $W_O$ was empirically determined. The best method for alignment for each dataset was chosen empirically. See Supplementary Table 10 for the parameters.

Finally, the closest points in the template to each centroid were converted into a binary string using the knowledge of the template. In the case of repetition datasets, if any of the binding sites corresponding to a single bit were activated, then the bit was set to 1. The decimal value of this binary string could then be converted using the alphabet encoding.

### 3D template alignment

Using the same general idea in 2D template alignment, Euclidean coordinates of centroids and the 3D cuboctahedron template were used to minimize the Euclidean squared distances of each centroid to the closest point on the template using the formula:

$$C = \sum_{i=1}^{n} (|A_i - B_i|)^2 \tag{2}$$

Initially, the 3D centroids and the 3D template were brought close by aligning the center of mass of the 3D centroids and the 3D template. The template was then $z$-scaled to fix the $z$-scale discrepancy due to centroid assignment by K-means clustering. Then, the minimization was done through a series of steps of x and y translation, then rotation about the $z$-axis with a total of 360°. Z translation could also be

performed, but it would cause the running time to be longer. The $z$ translation did not significantly affect the alignment result due to the small difference in the z value of the template and pattern center of mass. In all steps, the $C$ was calculated and the minimum value was taken for the final transformation. Following the alignment process, origamis that lacked at least one orientation marker were discarded, and the remaining results were analyzed. The process of converting binary was carried out in a similar manner to the 2D method, utilizing aligned patterns and the understanding of binary translation employed in the rules for pattern encryption.

### oxDNA simulation and 3D alignment of the unrelaxed and mean structure with experimental DNA-PAINT data

The simulation was carried out at 25 °C in a periodic boundary cube of length around 168 nm using oxDNA2 forcefield and Langevin thermostat, for $1.52 \times 10^{-5}$ s. An average of 6 such replicas were considered for the study, accounting for a total simulation time of $9 \times 10^{-5}$ s.

The DNA-PAINT data was projected onto the 2D $x$–$y$ plane and compared with simulation results to determine the best possible 2D configuration. The optimal 2D configuration conserves the $z$-axis and is subsequently scaled with a small factor to obtain a desirable structure. To obtain the best 2D configuration, the average configuration from the simulation was rotated such that the plane formed by the biotin strands was at the bottom of the frame and the normal of the plane was perpendicular to the $xy$ plane, similar to the experimental setup. The center of mass of the 12 docking handles was projected to the $x$–$y$ plane and fitted with the projection of the PAINT data points. Due to the geometry of the origami, when the structure is projected onto an $x$–$y$ plane, the 4 points at the extreme top along the $z$-axis and 4 at the very bottom overlap, making them indistinguishable from the clustering algorithm, causing them to look only as 8 clusters instead of 12. Using the k-means algorithm, the projected data points from the DNA-PAINT were clustered into 8 points instead of 12. The two configurations were fitted using the SVD superimposition technique (Fig. 6B for mean structure and Supplementary Fig. 13D for unrelaxed structure). The 2D configuration generated in this step remained unchanged throughout the later procedures.

With 2D alignment being fixed, the $z$-axis of PAINT data was increased by a small factor and clustered into 12 clusters. The sum of the RMSD between the centroids and the mean position of the docking handles was minimized for an optimal $z$-axis scaling factor (Supplementary Fig. 13E). To further confirm that the k-means produced meaningful clusters, the 12 centroids generated were again projected into the 2D plane, and the total RMSD between the current configuration and the old 2D configuration was noted. If the resultant RMSD exceeded the uncertainty, which was taken to be the RMSD from SVD 2D fitting from the previous step, the model was rejected. These results were visually confirmed as well to validate our findings. The k-means algorithm for the 3D fitting has an accuracy of 84%, which was predicted by executing the algorithm multiple times with random seeds and checking the projection 2D RMSD generated after 3D clustering (Supplementary Fig. 13E). With the final check of 2D RMSD, the algorithm was successfully able to figure out the scaling factor within a deviation of 6%.

### Statistics and reproducibility

No statistical methods were used to predetermine sample size. Within each region of interest (ROI) of DNA-PAINT super-resolution image, molecules were randomly selected by visual inspection, and only non-aggregating DNA origami structures were analyzed. For incorporation-efficiency analyses, the number of DNA origami samples denotes the number of non-aggregating origami per image. For clustering and template alignment, the sample size indicated in each figure caption was chosen based on technical feasibility (imaging time, storage, computation). For decryption readouts, 3 independent data-

processing runs (clustering, template alignment, and pattern matching) served as replicates to calculate means and estimate standard deviations. All AFM and DNA-PAINT experiments were independently repeated at least three times with consistent results. Investigators were not blinded to group allocation during experimentation or data analysis.

### Ethical statement

The methodology adopted for this study was reviewed and approved by the Institutional Biosafety Committee administered through Research Compliance Office at Arizona State University.

### Reporting summary

Further information on research design is available in the Nature Portfolio Reporting Summary linked to this article.

## Data availability

The source data in this study have been deposited in the repository[75] https://doi.org/10.5281/zenodo.17362995.

## Code availability

The scripts and codes for processing the data can be found in https://github.com/Jonathanzhao02/smlm_classification2d and https://github.com/gwisna/DNA-origami-cryptography-code-and-data.

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

## Acknowledgements

The research was funded by the National Science Foundation (NSF) through the SemiSynBio II (2027215) to H. Yan, C. Wang, and R. F. Hariadi and SemiSynBio III (2227650) to H. Yan, P. Šulc, and R.F. Hariadi. G.B.M. Wisna was supported by an American Heart Association (AHA) pre-doctoral fellowship (23PRE1029870). R.F. Hariadi acknowledges funding from the National Institutes of Health (NIH; 1DP2AI144247-01) and the NSF CAREER (2341002). P. Šulc acknowledges funding from the NSF CAREER (2239518). C. Wang acknowledges partial funding support from the NIH (1DP2GM149552). The authors thank Yinan Zhang, Pengkun Xia, Md Ashikur Rahman Laskar, Nirbhik Acharya, Ranjan Sasmal, Franky Djutanta, Ritvik Warrier, and Raymond Nucuta for their insightful discussions on our work. We also extend our thanks to Phillip R. Steen, Luciano A. Masullo, Hao Liu, and Raphael Jorand for the valuable discussions on 3D calibration and 3D DNA-PAINT and to Teun A.P.M. Huijben for his helpful insight into unsupervised classification. We also thank Tim Liedl and Gregor Posnjak for discussions and for providing Scaffold P8634 and 3D digital models of tetrapod DNA origami. Additionally, we thank Vaughn McGill-Adami for enabling ResNet.

## Author contributions

R.F.H., H.Y., C.W., and G.B.M.W. conceived the idea and designed the experiments. G.B.M.W., D.S. (Daria Sukhareva), and P.C. conducted the DNA origami, DNA-PAINT experiments, and analyzed the DNA-PAINT data. P.C. and D.S. (Deeksha Satyabola) performed AFM imaging. G.B.M.W. and J.Z. wrote the code for the clustering analysis for information readout and performed data analysis. S.R. and M.M. performed the oxDNA simulations and analyzed the simulation results under the supervision of P. Š. G.B.M.W., P.C., S.R. and R.F.H. contributed to the revision process. All authors read and edited the manuscript. R.F.H., H.Y., and C.W. secured the funding and supervised the work.

## Competing interests

The authors declare no competing interests.
