## [Transparent Peer Review file · Nature Communications]

High-speed 3D DNA PAINT and unsupervised clustering for unlocking 3D DNA origami cryptography

Corresponding Author: Professor Rizal Hariadi

Version 0:

Reviewer comments:

Reviewer #1

(Remarks to the Author)
Reviewer Summary

The authors present a DNA origami cryptography protocol that directly builds on the works of Zhang in 2019 (Nat Commun 10:5469) and Dickinson in 2021 (Nat. Commun 12(1):2371). In their approach, secure information storage was provided by combing a 3-step encryption technique with DNA-PAINT. Patterned docking strands on 2D DNA origami were first used for the encryption. The message was then transferred by mixing the DNA strands together with a ssDNA scaffold. The receiver unlocked the encryption and read the message only if the pattern rules were known. If the rules were unknown, at the time of the read-out, the message could not be retrieved and so, was deemed secure. The authors applied this approach to 3D DNA origami for the first time, demonstrating data encryption with 3D DNA-PAINT.

The results were well supported with proper controls, effective analysis, and sound microscopy. However, the use of DNA origami platforms to encrypt digital information was not original. The same basic principle was published in Nature Communications in 2019 by Zhang et al (see above). While the 2D DNA origami work is not original, its 3D counterpart was. More specifically, the 3D DNA-PAINT imaging on the 3D DNA wireframe was well executed and it creates a pathway towards data encryption on 3D DNA origami for the first time. That being said, the 3D DNA origami work lacks a wider and more complete analysis of the deeper features and characteristics that could be made possible with this technique. In response, I suggest publication in Nature Comm only after a full revision of the manuscript. What follows are comments in support of this process and the authors.

Nature Communications Questions and Reviewer Comments

(1) What are the noteworthy results?

- the use of 3D DNA-PAINT as an imaging technique to read-out digital data encrypted in 3D patterns of attached DNA short strands. As far as I know, this is the first documented attempt;

- the efficiency analysis of the incorporation of docking strands on a 2D DNA origami platform. This can improve DNA data storage on DNA origami and DNA-PAINT imaging on DNA nanostructures.

(2) Will the work be of significance to the field and related fields? How does it compare to the established literature? If the work is not original, please provide relevant references.

The 3D DNA origami encryption method was original and can influence the molecular data storage field. However, the encryption method on the 2D DNA origami was similar to the published Zhang et al. in 2019 where the pattern read-out was performed using AFM. Here instead, the authors used high-speed DNA-PAINT to read-out the message, which was also similar to the work of Dickinson et al. in 2021.

(3) Does the work support the conclusions and claims, or is additional evidence needed?
The majority of the claims are well supported by the results (see comments).

(4) Are there any flaws in the data analysis, interpretation and conclusions? - Do these prohibit publication or require revision?

I have not found any flaws in the interpretation of the results. However, a full revision is needed to align the claims and conclusions to the demonstration reported here.

(5) Is the methodology sound? Does the work meet the expected standards in your field?

Yes.

(6) Is there enough detail provided in the methods for the work to be reproduced?

Yes.

Major Revisions

- The authors presented the three steps of encryption as three keys without which the message cannot be retrieved. I find the use of the word "key" misleading in a few occasions. "Key" as in a secret rule to unlock the encryption, is not what I would call the "folding path" step: the m13mp18 scaffold sequence is the most common DNA scaffold sequence in the field; therefore, anyone could try to fold the cipher mix using the correct scaffold. I would suggest that the authors comment on how orthogonal custom-made scaffold sequences, applied to the presented pattern rules, can increase the encryption combinations.

- The authors well addressed the challenges of having a 3D DNA wireframe to read-out the data. This is mainly because of its flexibility and consequent variability of 3D positioning of the clusters' centroids. The manuscript will increase its value if the analysis on other 3D DNA origami shapes were taken into account. In particular it would be better to know if more rigid 3D structures can improve the resolution and if larger shapes can accommodate more data.

- Since the data encryption in 3D DNA origami is the main field advancement presented in this manuscript, I suggest the authors provide an analysis on the possible boundaries for this encryption. In particular this study is missing a demonstration of what are the advancements of using 3D DNA origami for encryption and what are the physical limits. This work presents an encryption of 4 characters using the 3D wireframe origami. Representative questions that should be investigated and answered are: How many characters can we encrypt at the maximum? Are other 3D DNA origami shapes better to go over 4 characters? What is the maximum height at which a 3D DNA origami that can be read with this method? Is the use of longer scaffolds advantageous or an intrinsic limitation?

- The authors explained clearly how high-speed DNA-PAINT is advantageous respect to the use of AFM to read-out patterns. This is well known in the field and the authors focused on this aspect to demonstrate the improvement in comparison to the work of Zhang et al. where AFM was used to read-out the data. I suggest the authors provide an analysis also on how to improve the data density encrypted in a single DNA origami rectangle. The available bits for the characters/letter are maximum 16 using the 48 matrix density (as depicted in fig.S14). This translates to 2 bytes of data per origami structure – a limit already demonstrated by Dickinson et al. A demonstration of higher data density per origami is essential to sustain the novelty and the claims of this study.

- Although extremely difficult, the docking strand can be visualized without knowing its sequence. This can be done exploiting the spurious binding of non-fully complementary sequences. Can the authors comment on the possible mechanisms to improve the orthogonality of the imager strand increasing its selectivity?

- I suggest the authors define more clearly detection efficiency and accuracy. Is it imaging accuracy? Data retrieving accuracy? About this on line 59 I would change "precision" with "accuracy".

- Define better what "information strand binder" is.

Minor Revisions

- Check the consistency between Cypher and Cipher throughout the manuscript and SI.

- It was confusing, for this reviewer, for figure 1B to be next door to 1C. For example, the average DNA-PAINT imaging on 1C (bottom) seems like the results from reading the wireframe 3D mix in 1B(bottom). It would help if the authors separated 1B and 1C with a line or equivalent.

(Remarks on code availability)

Reviewer #2

(Remarks to the Author)

The manuscript titled “High-speed 3D DNA PAINT and unsupervised clustering for unlocking 3D DNA origami cryptography” demonstrates an approach to DNA origami cryptography in which information is encoded and encrypted into a physical cypher mix through three components – templated pattern encryption, DNA origami scaffold routing, and DNA-PAINT imager strand sequence. Decryption of the cypher mix is performed by adding a specific staple strand folding mixture (key 3) to the cypher mix to form the full origami structures, reading the topological information from the DNA origami using DNA-PAINT with a unique imager strand (key 2), and then decoding the topological information using the appropriate pattern rules (key 1). The main advantage of this strategy is the use of the large key space of the origami scaffold routing pathway (>700 bits), though the advent of accessible DNA sequencing strategies reduces the effectiveness of this strategy against an attacker with sufficient motivation and resources.

The origami encryption strategy used in this work was demonstrated previously in reference 31 (<https://doi.org/10.1038/s41467-019-13517-3>). The only significant differences were adapting it for DNA-PAINT readout and developing strategies for binary data readout and error-correction from DNA-PAINT images, also previously demonstrated (<https://doi.org/10.1038/s41467-021-22277-y>). Encryption in 3D DNA origami (for which the work is named) represented only a small fraction of the work as proof-of-concept, and it was not clearly communicated what improvements it offered besides increasing the difficulty of readout. While the work is of high quality, the practicality of the method has yet to be demonstrated, and significant work is needed to show that the cyphers could be scaled up sufficiently to encode actual messages. Regardless of whether this work is published in Nature Communications, the authors would benefit by addressing the major concerns listed below.

Major concerns:

While the authors claim that 3 keys are needed to decrypt the messages, key 2 (DNA-PAINT imager sequence) is not a secure key as it can be determined directly from the cypher mix by nanopore sequencing. DNA sequencing is now fast, easy, and cheap, so this possibility cannot be overlooked.

Similarly, the key size of the folding path within the cypher mix itself is reduced by the presence of the docking site strands, and this reduction scales with the number of docking sites incorporated into the structure. Scaling up either the information density (more DNA-PAINT sites) or the message size (more unique origami) would further reduce the total key size for scaffold routing, and this information could be gathered directly from the cypher mix by sequencing.

Lesser concerns:

The motivations described in the first paragraph of the introduction should be revised to ensure clarity. For example, the authors mention silicon as being a material of limited availability, though I believe they mean semiconductor devices rather than silicon. DNA-based strategies are also not often considered viable substitutes for semiconductor-based devices in cases where availability would be an issue. Uses in niche cases, such as archival storage and physical encryption, are more likely to see DNA-based applications.

(Remarks on code availability)

Reviewer #3

(Remarks to the Author)

Wisna et al. demonstrate that DNA origami can be read out with super resolution microscopy for data storage applications. The authors present a strategy for data encryption that focuses on a practical implementation of a powerful encryption mechanisms and does not push the limits of any of the techniques. All data in the manuscript are presented clearly and the procedures and technical details reach the quality expected for Nature Communications. The discussion of the results and the use of a 3D strategy to store data in DNA are both well established. However, the combination with DNA Paint and single molecule based readout and advanced data technology are very well discussed.

Before publication the following minor changes are advised:

1) On page 1 line 22 the authors suggest that there is a shortage of silicon. This statements is obviously false as silicon is with almost 28% the second most abundant element in the earth crust by mass and is the second most abundant (oxygen is first). What they meant to say is probably that silicon-based chips were in short supply not silicon. This is an important difference that needs to be clarified.

2) The authors discuss the single molecule readout with AFM on page 2. However, DNA origami structures can be rapidly read with nanopore measurements as demonstrated a while ago Chen et al. Nano Letters 19(2):1210-1215, 2018 and also

reviewed in A. Doricchi et al. ACS nano, 16(11):17552-17571, 2022. The authors should mention that there are other - faster - possibilities to read data stored in the 3D structure of DNA.

3) Also on page 2 the authors state that they have a "fast" 24min read time at 10 nm resolution. Again, this comparison neglects that nanopores can reach similar resolution in arguably shorter time frames for the read out of information. See Chen. et al . Advanced Materials, 35(12): 2207434, 2023. It is fine to state that this is fast but should be clarified that this is only fast compared to AFM. Again, the focus on AFM as the other readout technique is only due to the limitations for reading out the structures that are made in this particular way.

4) It is a bit unclear if the authors image their three letter mixtures like "ASU" simultaneously or sequentially on the system. This should be clarified in the protocols and the main manuscript. Currently the authors state on page 19 that "Briefly, DNA origamis were immobilized on a BSA-biotin-streptavidin coated coverslip that forms a flow chamber with microscope slide glass through double-sided Kapton tape." Do they apply each origami separately or at the same time? (In Figure 3A it remains unclear if this is one design or several?)

5) The authors should comment if there are limits to the multiplexing readout in their method. If they have millions of letters (as suggested in the discussion), is it realistic that DNA PAINT can readout the different patterns in the same way?

6) Are there any limitations for the amount of DNA origami and material that has to be made to store large amounts of data? Did the authors consider how many molecules (including DNA oligos) they would need to store billions of bits.

(Remarks on code availability)

I believe that peer review process needs to stay anonymous to ensure the integrity.

Reviewer #4

(Remarks to the Author)

(Remarks on code availability)

Version 1:

Reviewer comments:

Reviewer #1

(Remarks to the Author)

The manuscript reports the first practical demonstration of 3D DNA-origami cryptography read out by high-speed 3D DNA-PAINT with unsupervised clustering, including controls and a clear analysis of detection efficiency and data-retrieval accuracy (≈ 70 – 89%). Noteworthy advances include extending earlier 2D encryption schemes (e.g., Zhang 2019; Dickinson 2021) into 3D, showing that structural rigidity (tunnel and tetrapod designs) restores expected spatial readouts, and articulating the cryptographic role of scaffold-routing key space with discussion of multi-orthogonal/custom scaffolds. The added treatment of physical limits (evanescent depth, capacity estimates) and the sequencing/attack considerations strengthens the work's framing. Methods, figures, and supplemental materials now provide sufficient detail for reproduction (sequences, buffers, imaging conditions, analysis workflow), and the methodology and analysis are sound. The main remaining weakness is practical scalability: per-origami payload is still demonstrated at ~ 2 bytes (with simulated paths to 4 bytes/multicolor) and high-throughput readout remains a bottleneck; a concise quantitative comparison table (AFM vs DNA-PAINT vs nanopore throughput/resolution) and release of analysis code/parameters and design files would further improve transparency. Overall, the data support the central claims, the contribution is significant to DNA nanotechnology, molecular cryptography, and super-resolution readout, and I recommend publication pending minor revisions to strengthen benchmarking and data/software availability.

(Remarks on code availability)

Reviewer #2

(Remarks to the Author)

The authors have sufficiently addressed my concerns, and the work is now suitable for publication in Nature Communications.

(Remarks on code availability)

I did not attempt to run the code, but the README file is highly instructive and the majority of the code is a replication of prior work.

Reviewer #3

(Remarks to the Author)

The authors improved the manuscript and addressed the comments. The challenge for fast readout remains due to the clearly stated need for many imaging cycles. While practical applications might be challenging - the concept is interesting and the high-resolution readout make this suitable for NComms.

(Remarks on code availability)

Reviewer #4

(Remarks to the Author)

(Remarks on code availability)

1 Reviewer #1 (Remarks to the Author)

Reviewer Summary

The authors present a DNA origami cryptography protocol that directly builds on the works of Zhang in 2019 (Nat Commun 10:5469) and Dickinson in 2021 (Nat. Commun 12(1):2371). In their approach, secure information storage was provided by combing a 3-step encryption technique with DNA-PAINT. Patterned docking strands on 2D DNA origami were first used for the encryption. The message was then transferred by mixing the DNA strands together with a ssDNA scaffold. The receiver unlocked the encryption and read the message only if the pattern rules were known. If the rules were unknown, at the time of the read-out, the message could not be retrieved and so, was deemed secure. The authors applied this approach to 3D DNA origami for the first time, demonstrating data encryption with 3D DNA-PAINT.

The results were well supported with proper controls, effective analysis, and sound microscopy. However, the use of DNA origami platforms to encrypt digital information was not original. The same basic principle was published in Nature Communications in 2019 by Zhang et al (see above). While the 2D DNA origami work is not original, its 3D counterpart was. More specifically, the 3D DNA-PAINT imaging on the 3D DNA wireframe was well executed and it creates a pathway towards data encryption on 3D DNA origami for the first time. That being said, the 3D DNA origami work lacks a wider and more complete analysis of the deeper features and characteristics that could be made possible with this technique. In response, I suggest publication in Nature Comm only after a full revision of the manuscript. What follows are comments in support of this process and the authors.

Major Revisions

- The authors presented the three steps of encryption as three keys without which the message cannot be retrieved. I find the use of the word “key” misleading in a few occasions. “Key” as in a secret rule to unlock the encryption, is not what I would call the “folding path” step: the m13mp18 scaffold sequence is the most common DNA scaffold sequence in the field; therefore, anyone could try to fold the cipher mix using the correct scaffold. I would suggest that the authors comment on how orthogonal custom-made scaffold sequences, applied to the presented pattern rules, can increase the encryption combinations.*

We thank the Reviewer for raising an important query. The cryptographic security in our approach that Reviewer 1 refers to does not rely on the sequence of the scaffold strand itself, but rather on the specific folding pathway by which the scaffold strand is routed to form the final DNA origami structure. While the circular M13mp18 scaffold sequence (7,249 nucleotides) is indeed publicly available, knowledge of this sequence alone is insufficient to determine the correct folding pathway required for successful DNA-PAINT readout. The folding pathway represents the spatial routing of a scaffold through a self-avoiding path that defines the precise 3D architecture of the origami structure. Even with complete knowledge

of the scaffold sequence, an adversary would face the computationally intractable challenge of determining the correct folding pathway from among an astronomically large number of possible self-avoiding configurations. This mathematical complexity, rather than sequence secrecy, provides the cryptographic security of our system.

In our study, we define folding paths as unique ways to route the DNA scaffolds used here (M13mp18 and P8634), in a self-avoiding path to form the DNA origami structure. Building on the foundational work by Zhang *et al.* on DNA origami encryption (<https://doi.org/10.1038/s41467-019-13517-3>) we adopt their scaffold folding strategy as a way to secure information. While Zhang *et al.* predicted that the number of possible folding pathways can create a key size of over 700 bits through some assumptions for 2D origami structure and a mathematical approach, see their supplementary information, the enumeration of self-avoiding polygons according to perimeter or area is actually still an unsolved problem as shown by Bousquet-Melou *et al.* (<https://doi.org/10.1007/BF01608785>, page 7). Therefore, predicting the folding paths/routings of the scaffold used to encrypt the information is extremely challenging for adversaries. For these reasons, using folding paths as a key for data encryption with DNA origami is justified appropriately. We have incorporated some of the argument in the revised introduction of paragraph 2, lines 34–43 of the manuscript, as shown below.

The central principle of modern cryptography, which employs complex mathematical problems to generate large possibilities of keys,¹ can be seamlessly translated into the DNA-origami system.² Zhang *et al.* estimated with some mathematical simplification that the key space resulted from self-avoiding scaffold foldings can exceed 700 bits with a DNA scaffold of 7249 nucleotides forming 2D DNA origami, as found in the M13mp18, potentially surpassing the Advanced Encryption Standard (AES) by a factor of >2 .³ However, enumeration of self-avoiding polygon according to perimeter or area is actually still an unsolved problem, indicating an astronomically large number of folding paths which goes beyond 700 bits.⁴ This will provide an opportunity to have a bigger key size to have more secure encryption if we can utilize the folding paths to form not only 2D but also 3D DNA origami.

We also discuss the application of multiple orthogonal scaffolds (lines 397–404) for assembling more complex and larger DNA origami structures, thereby increasing the total number of information strands that can be incorporated. However, we did not specifically mention custom-made scaffold sequences, as suggested by the Reviewer, and have modified the Discussion section to accommodate this suggestion. The modified sentences are shown below and implemented in lines 408–415.

Since the key size depends on the length of the scaffold, another possible avenue to increase the key size is to use several orthogonal scaffolds with custom-made sequences, thus enabling larger DNA origami. Multi-orthogonal scaffolding of larger DNA origami has been demonstrated.⁵ As an implication, with larger DNA origami and the same information density per origami, more information strands can be incorporated, thus giving a greater total information that can be encrypted in a specific DNA origami of multiple orthogonal

scaffolds. Using unique custom orthogonal sequences can further enhance security, as these sequences are different from commercially available scaffolds, which are derived from the M13mp18 scaffold.

- *The authors well addressed the challenges of having a 3D DNA wireframe to read-out the data. This is mainly because of its flexibility and consequent variability of 3D positioning of the clusters' centroids. The manuscript will increase its value if the analysis of other 3D DNA origami shapes were taken into account. In particular, it would be better to know if more rigid 3D structures can improve the resolution and if larger shapes can accommodate more data.*

We appreciate the Reviewer's insightful comment regarding the influence of DNA origami geometry on resolution and data capacity. To address Reviewer 1's question, we first used tunnel 3D DNA origami, and its analysis is presented in the revised supplementary information (Fig. S14). We demonstrated that using rigid 3D tunnel DNA origami, as illustrated by the oxDNA simulation (Fig. S14, bottom left panel) yielded a 3D DNA-PAINT result that aligned with the two docking separations depicted in the oxDNA simulation (Fig. S14, right panel). Thus, it is clear that the cuboctahedron's flexibility is the main reason we could not achieve agreement between the cuboctahedron design and the 3D DNA-PAINT result. We have added a new paragraph in the Results section explaining this result in lines 347–354.

We further performed control experiments using a more rigid 3D DNA origami tunnel structure (Fig. S14 and Table S6).⁶ The results from the DNA-PAINT experiments with DNA origami tunnels showed that the measured distances between dockings along the z -axis (~ 73 nm), closely matched the expected 75 nm separation as predicted by oxDNA simulations (Fig. S14). These findings indicate that the observed discrepancy in the dimensions of the 3D DNA-PAINT images of DNA origami cuboctahedrons along the z direction was due to structural flexibility, rather than to any limitations inherent in the 3D DNA PAINT imaging process itself. These results demonstrated that rigid 3D structures are essential for achieving accurate spatial readout in DNA origami cryptography.

To investigate larger rigid structures, we also investigated another rigid 3D origami tetrapod, designed by Posnjak et al. (<https://www.science.org/doi/10.1126/science.ad12733>) which has ~ 35 nm arm length. We demonstrated a 35 nm resolution in x , y , and z planes (new Fig. 7A and B) in agreement with the design and AFM images (new Fig. 7A and C). The results further support our finding that cuboctahedron flexibility is the reason for the mismatch between 3D DNA-PAINT results and the design along the z direction. To demonstrate incorporation of higher data points, we constructed a dimer by joining two tetrapods by their ends without any rotation and placed docking sites to encode higher data density as shown in new Fig. 7D,E. Such dimer assembly show that a larger structure is able to accommodate more sites (8 sites) to place the information strands for encryption as compared to a monomer (4 sites). Together, these results confirm that increased rigidity in 3D DNA origami designs leads to improved resolution in DNA-PAINT readouts, and that larger, modular assemblies can successfully accommodate higher data densities. We added a paragraph in the Results section explaining this finding in lines 355–375 and provided Figure 7 as an additional Figure in the main text of the revised manuscript.

Encrypting information in a multimer structure with a rigid 3D DNA origami tetrapod dimer. To address the potential resolution limitations caused by structural flexibility, we demonstrated the encryption of information with using a rigid 3D DNA origami structure known as a tetrapod (Fig. 7 and Table S7).⁷ The rigidity of the tetrapods arises from their 24-helix thick arms, in contrast to the 2-helix arm of the DNA origami cuboctahedron in Fig. 5. To assess rigidity of tetrapods, we placed 4 docking strands at the end of each tetrapod arm (Fig. 7A, top left panel) and performed 3D DNA PAINT imaging. Three-dimensional DNA-PAINT images showed that the measured distances between the 4 docking spots in both the xy plane and the z direction match the design (Fig. 7B). To encrypt information, we performed dimerization of the tetrapod through one of the arms by connecting the sticky ends (Fig. 7D) and constructed the encryption rule for the pattern, comprising number bits, position bits, and alignment markers (Fig. 7E). Dimerization was required to increase the information capacity beyond 4 bits, thereby enabling the encryption shown in Fig. 7E. We encrypted the information “1776” (Fig. 7G, top panel, and Table S8 for the mixing of strands to form the monomer corresponding to the pattern), so that combining cuboctahedron and dimer tetrapod encryption encodes the information for July 4th, 1776. As a proof of concept, we performed the experiment by individually synthesizing and purifying 2 monomers corresponding to each specific pattern prior to dimerization of the DNA-PAINT sample. We selected the best single molecule for each dimer pattern from the DNA-PAINT experiment, where the experimental results match the designed pattern (Fig. 7G, bottom panel, and Fig. S15, showing the rest of the selections). Dimerization added another layer of security to form the correct pattern in which information is encrypted; a correct dimer must form. These results validated that polymerization can be used to increase the number of bits, allowing more characters to be encrypted and adding an extra security layer.³

- *Since the data encryption in 3D DNA origami is the main field advancement presented in this manuscript, I suggest the authors provide an analysis on the possible boundaries for this encryption. In particular this study is missing a demonstration of what are the advancements of using 3D DNA origami for encryption and what are the physical limits. This work presents an encryption of 4 characters using the 3D wireframe origami. Representative questions that should be investigated and answered are: How many characters can we encrypt at the maximum? Are other 3D DNA origami shapes better to go over 4 characters? What is the maximum height at which a 3D DNA origami that can be read with this method? Is the use of longer scaffolds advantageous or an intrinsic limitation?*

We thank the Reviewer’s assessment in identifying areas for improvement. We have provided explanations in the main text highlighting the role of complex of 3D DNA origami geometries with data encryption. Considering the limitations in pattern recognition with 2D imaging modalities (as shown in Fig. S10), we have modified the introduction and result sections as addressed in the response to the first comment. Revised text can be located in section titled **The 3D wireframe DNA origami encryption, decryption, and fast readout**. We have modified the main text with the following version to discuss possible analysis boundaries for the 3D encryption, lines 286–306.

Implementing DNA origami cryptography in 3D nanostructures enhances security by firstly expanding the unused folding paths key space if only 2D origami is being used to encrypt the information, and secondly concealing the encrypted pattern within complex 3D architectures, as opposed to the prior approach using 2D origami nanostructures.³ Several groups have demonstrated various 3D DNA origami shapes, including the cuboctahedron (Figs. 1B and 5), which has a radius of ~ 35 nm.^{8,9,9-12} Revealing the true shape of and the encrypted pattern on 3D DNA origami required 3D super-resolution imaging with sub-25 nm resolution. In comparison, AFM, which is restricted to 2D imaging, may misinterpret the 3D wireframe cuboctahedron as a 2D construct, leading to potential confusion about the encrypted pattern (Figs. 1B and S10). We identified the high-speed DNA-PAINT, capable of imaging 3D patterns on 3D DNA origami with ≤ 10 nm z -resolution,^{6,13,14} as a superior readout technique for 3D DNA origami cryptography. To demonstrate DNA origami cryptography in 3D, we encrypted the date “0407” (4th of July) on a 3D wireframe DNA origami cuboctahedron template. In our design, the docking/information strand binding length to the scaffold ranges from 52–54-nt (Table S2). The pattern encryption allocated 4 bits (ABCD) for number encryption, 3 bits (EFG) for position encryption, allowing a maximum of 8 characters being encrypted, and 5 bits for alignment markers (3 filled and 2 empty) to break the in-plane (xy) rotational symmetry (Fig. 5A). The bits are located at the vertices of the cuboctahedron, with the possibility of placing more bits on the edges. The distance between 2 stacking bits along the z axis is 65–70 nm based on the ideal design (Fig. S11). We acquired 43,510 frames with an exposure time of 50 ms, resulting in a total acquisition time of ~ 35 mins. We then selected 150–210 3D DNA origami structures for each pattern using visual inspection (Fig. S12).

Additionally, to highlight the analysis on the general physical limit on 3D DNA origami encryption, the following modifications are made in lines 458–475.

In the case of 3D DNA origami, since DNA-PAINT requires TIRF, which employs evanescent waves to obtain high-resolution reconstructions, the limit of the 3D origami height that can be imaged follows the decay length of the evanescent wave, which is around 150–200 nm from the surface. In an ideal case with the state-of-the-art DNA-PAINT that can achieve around 1 nm resolution in x , y , and z resolution,¹³ we can estimate that with an ideally rigid 3D structure of a tunnel shape that can be formed with a single scaffold with a typical length of 8000 nt. The origami will have a height of 75 nm and side lengths of 16 nm, and we can design a 5 nm separation in z direction and 2.5 nm in x and y directions for the information strands. Therefore, we can accommodate around 15 bits along the z direction with a height of 75 nm and 20 bits along the x and y directions. In total, we can accommodate 15×20 bits, which is equal to 300 bits for a single origami formed from a single scaffold. The 300 bits can then be distributed for a specific number of alignment markers, bits for characters, and position encryptions. A particular example case for the distribution of these 300 bits is 10 bits for alignment markers, 8 bits for the ASCII, and adding 2 redundancies totaling 24 bits, and the remaining 266 bits can be used for position encryption, which is enough to encrypt 2^{266} characters. This particular simple analysis with ideal assumptions was performed to show the physical limitations on the total number of characters that can be encrypted, assuming that we can achieve ideal DNA-PAINT

super-resolution imaging and structural rigidity conditions using specific 3D DNA origami. Longer scaffolds can be exploited to create a larger origami in the x , y , or z directions, thus giving more bits along those directions.

- *The authors explained clearly how high-speed DNA-PAINT is advantageous respect to the use of AFM to read-out patterns. This is well known in the field and the authors focused on this aspect to demonstrate the improvement in comparison to the work of Zhang et al. where AFM was used to read-out the data. I suggest the authors provide an analysis also on how to improve the data density encrypted in a single DNA origami rectangle. The available bits for the characters/letter are maximum 16 using the 48 matrix density (as depicted in Fig. S14). This translates to 2 bytes of data per origami structure – a limit already demonstrated by Dickinson et al. A demonstration of higher data density per origami is essential to sustain the novelty and the claims of this study.*

We thank the Reviewer for this comment. We want to point to the study performed by Dickinson et al. that uses rectangular origami with 8 by 6 matrix. They distribute 48 bits into 16 bits=2 bytes for droplet, 4 bits for checksum, 4 bits for index, 20 bits for parity checks and 4 bits for orientation marker that translate to 2 bytes of data per origami structure. Similarly, we have demonstrated an encryption on a 2D origami with a rectangular origami with 48 total bits and 2 bytes of data per origami in the main text Figure 5. While increasing the information density per origami is advantageous, it is beyond the scope of this study. This study focuses on exploiting the astronomical number of folding paths for scaffold DNA to form 2D and 3D origami to enhance encryption security significantly, as addressed in the previous comments. We also focus on incorporating high-speed DNA-PAINT to improve data acquisition speed up to $6\times$ as compared to the work by Dickinson et al. Moreover, 2 bytes per origami is already sufficient to encrypt many data types such as char, bool, unsigned _int16, _int16, unsigned _int8, _int8, signed char, unsigned char, short, unsigned short, wchar_t following Microsoft's nomenclature (<https://learn.microsoft.com/en-us/cpp/cpp/data-type-ranges?view=msvc-170>).

Higher data density per origami can be achieved with more bits per origami. One way is to have bits separated with a 5 nm distance, therefore doubling the number of bits per origami to be 96 in total. Further, two-color DNA-PAINT imaging can be used to increase information content by keeping the distance of 10 nm between bits as proposed in Fig. S16. We can also vary the layout for encryption to accommodate more bits for the character bits while decreasing the number of bits for position or index. Next, raw DNA-PAINT data can be simulated to generate patterns for the above approaches with Picasso Design and the Simulate platform developed by the Jungmann lab (<https://doi.org/10.1038/nprot.2017.024>). We present simulated datasets in Fig. S17 as we acknowledge that our microscope setup is already reaching the limit with 10 nm resolution, as shown by the data in Figs. 2 and 4. Corresponding main text modifications to explain these strategies in the discussion section lines 446–457.

Another notable demonstration is the improvement in data density per origami. Our current experimental realization has the same docking density of 48 dockings per origami as in the work by Dickinson et al.,¹⁵ which translates to a maximum of 2 bytes per origami

data density. This density is already sufficient to encrypt many data types, such as char, bool, unsigned int16, int16, unsigned int8, int8, signed char, unsigned char, short, unsigned short, wchar_t, all of which occupy only 1 or 2 bytes for one value.¹⁶ However, increasing the data density per origami is also useful in the case of encryption of several data types, such as int or float, which occupy 4 bytes for a value. To increase the data density per origami, in addition to the two-fluorophore strategy mentioned above with slight modification of the encryption pattern layout to achieve 4 bytes per origami data density, making the separation between docking spots smaller than 5 nm also results in more docking spots per origami, thus achieving a higher data density per origami of 4 bytes per origami. We showed the simulated DNA-PAINT using the Picasso Simulate module¹⁷ to demonstrate the possibility of a data density of 4 bytes per origami (Fig. S17).

- *Although extremely difficult, the docking strand can be visualized without knowing its sequence. This can be done exploiting the spurious binding of non-fully complementary sequences. Can the authors comment on the possible mechanisms to improve the orthogonality of the imager strand increasing its selectivity?*

We agree with the Reviewer's comment on the difficulty of visualizing the docking strand with spurious interaction. We believe spurious interactions will produce faint binding signals because it is very fast and will not yield the required 10 nm resolution ($20\times$ below the diffraction limit), leading to failure in image reconstruction. Consequently, the images will be blurry, and it will be almost impossible to recover the unique pattern on the origami. Therefore, adversaries trying to use this method to crack the pattern will essentially not be successful. We have added a sentence to describe this in the result section, lines 130–134.

In the case where the adversaries tried to use spurious interaction to image the docking, it would only result in blurry pattern due to short binding of spurious interaction or even if they could obtain the exact docking sequence through DNA sequencing, the true pattern would not be revealed via DNA-PAINT super-resolution imaging because of the highly secured DNA origami encryption which is impossible to crack⁴ (Fig. 1C).

Docking sequence orthogonality plays an important role when utilizing multi-color DNA-PAINT to increase the data density per origami as discussed in the previously. It is essential to have the docking strands corresponding to fluorophore 1 to be orthogonal to the docking which corresponds to fluorophore 2. This reduces cross-hybridization between the two sequence leading to optimum imaging of the pattern on the DNA origami at super-resolution. We have added this description in the revised discussion section lines 422–431.

Second, to further increase information density, multicolor DNA-PAINT can be used.^{6,18} For instance, in the case of a 2D RRO template with 48 docking strands per origami, using 2 fluorophores such as ATTO 488 and ATTO 647 for 2 orthogonal high-speed docking sequences would allow for an additional 36 docking spots (assuming that 12 docking spots of the new color are also being dedicated for alignment markers). These docking spots can be used for 18 additional position bits and 18 redundancy bits, collectively resulting

in 28 position bits, capable of encoding ~ 268 million characters (Fig. S16). It should be noted that the orthogonality of the 2 imager strands labeled with different fluorophores has to be met to prevent spurious or cross-interactions between the imagers. The spurious interaction between imagers can negatively impact DNA-PAINT super-resolution imaging, manifested in the increased background, thus compromising resolution.

- *I suggest the authors define more clearly detection efficiency and accuracy. Is it imaging accuracy? Data retrieving accuracy? About this on line 59 I would change “precision” with “accuracy”*

We agree with the Reviewer’s comment and have changed the word “precision” with “accuracy” from the introduction section lines 73–76.

Integration of bit redundancy improves the accuracy in the retrieval of data from 2D and 3D DNA origami encryption and decryption processes, achieving an accuracy of 70–89%, despite the inherent flexibility of 3D DNA origami architectures, as predicted by computational modeling of oxDNA.

Regarding the detection efficiency. It is a measure of the ability to incorporate the information/docking strands on the origami and to be able to image and visualize them with high speed DNA-PAINT. The detection efficiency measures the number of information/docking strands that can be reconstructed by DNA-PAINT while the data retrieval accuracy measures in terms of how many correct encrypted characters are being retrieved. We added a definition on the detection efficiency in the result section lines 156–169.

Ensuring secure information transmission requires a high level of information retention and retrieval. Our encryption-decryption protocol relies on DNA origami architecture with integrated DNA-PAINT and unsupervised clustering readout. Previous work on DNA origami combined with DNA-PAINT readout for storage applications¹⁵ showed a high error rate, thus requiring a mechanism and algorithm for robust error correction for each origami data droplet. Therefore, it is crucial to assess the incorporation and detection efficiency of the information strands on the RRO origami template to achieve higher information retention and retrieval. The two metrics are the measures on the ability to incorporate the docking strands on the origami and to be able to image and visualize them with high-speed DNA-PAINT. We investigated the detection efficiency of information strands on the 2D RRO DNA origami template via DNA-PAINT, where 1 information strand would be translated into 1 docking spot. To this end, we varied the number of spots per origami between 12, 24, and 48 as shown in Figs. 2A and S2. The increasing number of spots per origami led to a reduction in the separation between spots. Specifically, the shortest distance between spots in origami with 12, 24, and 48 spots were systematically designed to be 20, 14, and 10 nm, respectively.

- *Define better what “information strand binder” is*

We have added a definition to improve the clarity on “information strand binder” as shown in the revised Fig. 2A caption and the revised result section lines 171–175.

We started with an RRO DNA origami containing 12 spots/ origami, with a 32-nt information strand binder (Fig. 2A(i)), and estimate detection efficiency following the method developed by Strauss et al.¹⁹ (Fig. S3). Information strand binder was a section in the information strands attached to the origami scaffold, thus facilitating placement of the information strand on a specific site on the origami (Fig. 2A, green-colored section).

Minor Revisions

- *Check the consistency between Cypher and Cipher throughout the manuscript and SI.*

We appreciate the Reviewer for pointing out minor issues. We have changed “Cypher” to “Cipher” throughout the main text and SI.

- *It was confusing, for this Reviewer, for Figure 1B to be next door to 1C. For example, the average DNA-PAINT imaging on 1C (bottom) seems like the results from reading the wire-frame 3D mix in 1B(bottom). It would help if the authors separated 1B and 1C with a line or equivalent.*

We thank the Reviewer for the suggestion and have widen the spacing between Fig. 1B and Fig. 1C so that it will be clear they are two different panels.

2 Reviewer #2 (Remarks to the Author)

Reviewer Summary

The manuscript titled “High-speed 3D DNA PAINT and unsupervised clustering for unlocking 3D DNA origami cryptography” demonstrates an approach to DNA origami cryptography in which information is encoded and encrypted into a physical cypher mix through three components – templated pattern encryption, DNA origami scaffold routing, and DNA-PAINT imager strand sequence. Decryption of the cypher mix is performed by adding a specific staple strand folding mixture (key 3) to the cypher mix to form the full origami structures, reading the topological information from the DNA origami using DNA-PAINT with a unique imager strand (key 2), and then decoding the topological information using the appropriate pattern rules (key 1). The main advantage of this strategy is the use of the large key space of the origami scaffold routing pathway (~700 bits), though the advent of accessible DNA sequencing strategies reduces the effectiveness of this strategy against an attacker with sufficient motivation and resources.

The origami encryption strategy used in this work was demonstrated previously in reference 31 (<https://doi.org/10.1038/s41467-019-13517-3>). The only significant differences were adapting it for DNA-PAINT readout and developing strategies for binary data readout and

error-correction from DNA-PAINT images, also previously demonstrated (<https://doi.org/10.1038/s41467-021-22277-y>). Encryption in 3D DNA origami (for which the work is named) represented only a small fraction of the work as proof-of-concept, and it was not clearly communicated what improvements it offered besides increasing the difficulty of readout. While the work is of high quality, the practicality of the method has yet to be demonstrated, and significant work is needed to show that the cyphers could be scaled up sufficiently to encode actual messages. Regardless of whether this work is published in Nature Communications, the authors would benefit by addressing the major concerns listed below.

Major concerns:

- *While the authors claim that 3 keys are needed to decrypt the messages, key 2 (DNA-PAINT imager sequence) is not a secure key as it can be determined directly from the cypher mix by nanopore sequencing. DNA sequencing is now fast, easy, and cheap, so this possibility cannot be overlooked.*

Similarly, the key size of the folding path within the cypher mix itself is reduced by the presence of the docking site strands, and this reduction scales with the number of docking sites incorporated into the structure. Scaling up either the information density (more DNA-PAINT sites) or the message size (more unique origami) would further reduce the total key size for scaffold routing, and this information could be gathered directly from the cypher mix by sequencing.

We thank the Reviewer for pointing out a valuable concern. We agree that with advancement of the DNA sequencing techniques data encryption with DNA origami will allow sophisticated encryption designs. However sequence identification alone does not provide full decryption key. Our approach relies on the combination of routing complexity and integration multiple keys (key 1 and 2) as stated in the result section. Key 1 and 2 are small in size with 84 bits and 16 bits. Key 3 is nearly impossible to crack as discussed in the reply to the first Reviewer's comment No. 5.

Next, we performed an experiment to demonstrate the need of complex folding path with all staple strands. The nanostructure was constructed by mixing the scaffold M13mp18 with docking sequences for the 10 nm grid (key 2), but without adding the rest of the staples (key 3). DNA-PAINT imaging of this nanostructure followed by reconstruction of the super-resolution image did not show a 10 nm grid pattern. This result is presented in Fig. S18 Therefore we demonstrated that the unique pattern was not visible by using only key 2 i.e. 48 information strands with 64 binder length that make a total base pairing of $48 \times 64 \text{ nt} = 3072 \text{ bp}$ with the scaffold out of a total scaffold length of 7249. This shows that without all keys, the correct shape does not form and the true pattern is not revealed, suggesting that a very unique of remaining staple strands set has to be applied. Subsequently, applying brute force to guess and try every possible folding paths is almost impossible. Even with the knowledge about correct shapes of origami and the pattern encryption rules (key 1), it is still impossible for the adversaries to estimate the actual folding path, see Fig S18 for the schematics to help visualizing the explanation.

We also want to point out that that Zhang et al. (<https://doi.org/10.1038/s41467-019-13517-3>) in their seminal work have argued on the possibility of DNA sequencing to crack the encryption where they wrote in their result section "In practice, the chance to replace

the DNA media by a counterfeit during delivery is little. Laborious sequencing is required to find out the length and sequence of the scaffold strand. After that, Mallory would have to crack the specific routing and sliding of the scaffold in the DNA origami using an exhaustive method. Any variation on either of the factors would result in a detectable variation of the pattern”. Therefore, we think that this concern is valid. While there is no encryption system that is not susceptible to hacking but with more than 700 bits of keysize which is equal to 2^{700} possible solutions, it will be almost impossible to crack the encryption with brute force methods. We discussed this in the revised discussion section lines 476–484.

We additionally conducted an experiment to highlight the necessity of a complete set of staple strands for proper folding. The nanostructure was assembled by combining the M13mp18 scaffold with docking sequences for the 10 nm grid (key 2), while omitting the remaining staples (key 3). DNA-PAINT imaging and subsequent super-resolution reconstruction revealed no 10 nm grid patterns (Fig. S18). This confirms that using only key 2 with 48 information strands and 64-nt binders, totaling 3072 bp of hybridization with the 7249-nt scaffold, was insufficient to generate the intended pattern. Thus, without the full complement of the staple strands, the correct structure and encoded pattern do not emerge. This finding underscores that the remaining staples must follow a highly specific set for successful folding. Consequently, attempting brute-force approaches to predict all the possible folding paths is practically infeasible.

Lesser concerns:

- *The motivations described in the first paragraph of the introduction should be revised to ensure clarity. For example, the authors mention silicon as being a material of limited availability, though I believe they mean semiconductor devices rather than silicon. DNA-based strategies are also not often considered viable substitutes for semiconductor-based devices in cases where availability would be an issue. Uses in niche cases, such as archival storage and physical encryption, are more likely to see DNA-based applications.*

We thank for the suggestion and agree with them regarding the difference between semiconductor materials and devices availability and the archival application of DNA-based storage. The revised section is in lines 17–27.

The development of semiconductor-based transistors in the mid-20th century marked the beginning of the information revolution, enabling data storage and computing, while the advent of fiber optics and optical amplifiers necessitated secure communication systems.²⁰ Modern cryptography, such as the Advanced Encryption System (AES) with keys up to 256 bits, evolved to meet this need, relying heavily on semiconductor devices for processing power.^{1,21,22} However, the limited availability of these semiconductor devices due to high demands in many advanced technologies with long lead time to manufacture,²³ and the high energy demand for operating devices²⁴ present significant challenges. DNA, with its stability, programmability, high information density, and low maintenance needs, has emerged as a promising substitute.^{25–28} Pioneering work on DNA computing^{29–35} and archival data storage^{15,36–38} has demonstrated DNA’s potential for these applications, making the de-

velopment of molecular cryptography protocols crucial for secure DNA-based archival data storage management in the future.

3 Reviewer #3 (Remarks to the Author)

Wisna et al. demonstrate that DNA origami can be read out with super-resolution microscopy for data storage applications. The authors present a strategy for data encryption that focuses on a practical implementation of powerful encryption mechanisms and does not push the limits of any of the techniques. All data in the manuscript are presented clearly and the procedures and technical details reach the quality expected for Nature Communications. The discussion of the results and the use of a 3D strategy to store data in DNA are both well established. However, the combination with DNA Paint and single molecule based readout and advanced data technology are very well discussed.

Before publication, the following minor changes are advised:

- On page 1 line 22 the authors suggest that there is a shortage of silicon. This statements is obviously false as silicon is with almost 28% the second most abundant element in the earth crust by mass and is the second most abundant (oxygen is first). What they meant to say is probably that silicon-based chips were in short supply not silicon. This is an important difference that needs to be clarified.*

We appreciate Reviewer 3's comment, as it is similar to Reviewer 2's comment and has been thoroughly addressed previously in this cover letter.

We acknowledge the comments provided by Reviewer 3, which align with those of Reviewer 2, and have been thoroughly addressed previously in this cover letter.

- The authors discuss the single molecule readout with AFM on page 2. However, DNA origami structures can be rapidly read with nanopore measurements as demonstrated a while ago Chen et al. Nano Letters 19(2):1210-1215, 2018 and also reviewed in A. Doricchi et al. ACS nano, 16(11):17552-17571, 2022. The authors should mention that there are other - faster - possibilities to read data stored in the 3D structure of DNA.*

We apologize for overlooking nanopore to read data stored in DNA molecules. The referred articles describe a method for reading DNA nanostructures attached to long duplex DNA. Although they do not use 2D and 3D DNA origami, we are inclined to agree with Reviewer 3 for using nanopore techniques for DNA nanostructure readout for data storage applications. We believe this complements high-speed DNA-PAINT imaging utilized in our manuscript. The revised introduction now discusses nanopore techniques for reading duplex DNA data strings with unique 3D DNA junctions used as barcoding labels to distinguish the encoding. We have cited several relevant papers, including the papers mentioned by Reviewer 3, related to these advancements in lines 44–65.

Despite the theoretical large key space, the practicality of the current demonstration of DNA origami is limited by the inherently slow imaging of conventional atomic force microscopy (AFM)³⁹ coupled with the reliance on biotin-streptavidin conjugations, which frequently result in undesired aggregations.^{40,41} Another way to read data stored in mostly tubular-shaped DNA duplex decorated with DNA nanostructures is by utilizing nanopore sequencing.^{42,43} Chen et al. and Zhu et al. have shown that multilevel encoding with different barcodes with DNA nanostructure multi-way junctions and dumbbell shapes can improve the data readout using nanopore.^{44,45} It can reach readout resolution of different nanostructure barcodes with separation up to 6 nm within tens of microsecond for one translocation event of duplex with length of around 100-200 nm.⁴⁴ However, due to the geometry of the nanopore and the readout process of the data which has to be in a sequential manner as the samples translocate the pore, the application for DNA origami data storage readout is limited to DNA nanostructures with tubular form. On the other hand, high-speed DNA-PAINT,⁴⁶ a DNA-based super-resolution imaging technique, offers a solution to enhance readout speed and eliminate aggregation induced by protein conjugation. This method capitalizes on the stochastic binding of single-stranded DNA (ssDNA), referred to as docking strand, and short fluorophore-labeled ssDNA called imager strands to achieve resolutions as high as sub-nm¹³ to 10 nm.¹⁸ DNA-PAINT enables faster imaging and a larger imaging area of nearly 100 μm by 100 μm , allowing for the imaging of thousands of complex shapes of 1D, 2D and 3D DNA origami nano structures per field of view.⁴⁶ Despite the application of DNA origami and DNA-PAINT techniques for alternative DNA storage, high error rates during experimental origami folding and low detection efficiency in DNA-PAINT imaging necessitate the development of error-correction post-processing algorithms.¹⁵ Therefore, optimizing these strategies is crucial for improving information retrieval in DNA-based systems.

- *Also on page 2 the authors state that they have a "fast" 24min read time at 10 nm resolution. Again, this comparison neglects that nanopores can reach similar resolution in arguably shorter time frames for the read out of information. See Chen. et al . Advanced Materials, 35(12): 2207434, 2023. It is fine to state that this is fast but should be clarified that this is only fast compared to AFM. Again, the focus on AFM as the other readout technique is only due to the limitations for reading out the structures that are made in this particular way.*

We thank the reviewer for the suggestion. We have simultaneously addressed this comment in the comment No. 2 above by discussing and citing the super-resolution nanopore readout of DNA nanostructure dumbbell attached to long duplex DNA which was referred by Reviewer 3.

- *It is a bit unclear if the authors image their three letter mixtures like "ASU" simultaneously or sequentially on the system. This should be clarified in the protocols and the main manuscript. Currently the authors state on page 19 that "Briefly, DNA origamis were immobilized on a BSA-biotin-streptavidin coated coverslip that forms a flow chamber with microscope slide glass through double-sided Kapton tape." Do they apply each origami separately or at the same time? (In Figure 3A it remains unclear if this is one design or several?)*

We thank the Reviewer for pointing out this. We have modified the methods to improve the

clarity of the experiment that we did for all the DNA origami patterns. The revised method section is in lines 700–715.

DNA-PAINT super-resolution imaging was performed following the detailed protocols previously described by Schnitzbauer et al.¹⁷ Briefly, DNA origamis with specific patterns such as “N”, “S”, “F” (Fig. 3), “A”, “S”, “U” (Fig. 4), “0”, “4”, “0”. “7” (Fig. 5) and “1”, “7”, “7”. “6” (Fig. 7) were immobilized on a different BSA-biotin-streptavidin coated coverslip that forms a flow chamber with microscope slide glass through double-sided Kapton tape. Buffer A+ (10 mM Tris-HCl, 100 mM NaCl, 0.05% (v/v) Tween 20, pH 8.0) was used to dilute BSA-Biotin and streptavidin to 1 mg/ml and 0.5 mg/ml concentration, respectively. Buffer B+ (5 mM Tris-HCl, 10 mM MgCl₂, 1 mM EDTA, 0.05% (v/v) Tween 20, pH 8.0) was used to dilute DNA origami at experimental concentrations. Buffer B+ was also used to dilute the imager strands (AGGAGGA/3' Cy3B/) to experimental concentrations. Oxygen scavenger solutions PCA, PCD and Trolox with final concentrations of 1.25X PCA, 1X PCD, and 1X Trolox were mixed with the imager strands to make the final imaging solution. Individual letter or pattern was imaged separately so that the accuracy study can be carried out by knowing that all the picked origamis from specific letter sample is coming from single pattern thus there were no false negatives and false positives which can interfere the data retrieval accuracy results. In the real scenario, mixing all patterns can be done to simplify the imaging steps. The experimental conditions for each figure are described in the Table S4.

- *The authors should comment if there are limits to the multiplexing readout in their method. If they have millions of letters (as suggested in the discussion), is it realistic that DNA PAINT can readout the different patterns in the same way?*

The Reviewer raises a valuable point regarding the feasibility of DNA PAINT for reading out millions of distinct patterns, as might be required for large-scale encoding tasks. While DNA PAINT provides high-resolution imaging to distinguish patterns of encrypted letters with high fidelity, the scalability to millions of letters or patterns in a single imaging session does present a practical challenge. We have discussed this in the revised discussion section lines 432–445.

A practical limitation in terms of readout has also to be noted. While the proposed origami design presented in Fig. S14 can accommodate millions of characters to be encoded, performing a readout of millions of characters is practically challenging in terms of the readout time. The challenge is to have enough origami to be imaged so that a reasonable number of origami for each pattern can be collected for decoding or decryption. With a global accuracy of around 75–89%, we would want to collect at least 100 origami images for each pattern. This means that for one million letters, we would need to acquire 100 million origami images. High-speed DNA-PAINT is a high-throughput method that can capture approximately 5000–10000 origami per field of view (500×700 μm²) in a single imaging session per microscope. Therefore, obtaining 100 million origami molecule images would require 10,000 imaging sessions, which is practically challenging in terms of the readout time. We would say the current practical limit is to perform 10–20 imaging sessions where

we can obtain a maximum 200,000 origami that can accommodate around 2,000 letters, and these imaging sessions would consume around 6–12 hours if a session takes 35 minutes, as we described in the manuscript. To scale it up, additional microscopes can be employed to expedite the process.

- *Are there any limitations for the amount of DNA origami and material that has to be made to store large amounts of data? Did the authors consider how many molecules (including DNA oligos) they would need to store billions of bits.*

We appreciate the Reviewer's thoughtful suggestion. The capacity of DNA origami for data storage is influenced by complex geometry and the number of staple strands required for assembly. We believe that a very large library of DNA origami will be needed when scaling to billions of bits. But advances in high-throughput DNA synthesis and assembly should make it feasible to produce strand pool. Moreover, DNA nanotechnology allows modular assembly of multiple DNA origami structures to enable high data density. Although our current study focuses on proof-of-concept demonstrations, we believe our approach paves the way for future studies that can evaluate practical scaling strategies.

4 Reviewer #3 (Remarks on code availability):

- *I believe that peer review process needs to stay anonymous to ensure the integrity.*

We followed Nature Communications submission guidelines and unfortunately it is not in our hands to control the anonymity.

5 Reviewer #4 (Remarks to the Author)

- *I co-reviewed this manuscript with one of the Reviewers who provided the listed reports. This is part of the Nature Communications initiative to facilitate training in peer review and to provide appropriate recognition for Early Career Researchers who co-review manuscripts.*

We are grateful to Reviewer 4 for their time in reviewing and evaluating our manuscript, as well as to Nature Communications for the initiative designed to support the training of early career researchers.

End of Responses to Referees

References

- ¹ Jonathan Katz and Yehuda Lindell. Introduction to modern cryptography. CRC press, 2020.
- ² Ashish Gehani, Thomas LaBean, and John Reif. DNA-based cryptography. In Aspects of Molecular Computing, Lecture notes in computer science, pages 167–188. Springer Berlin Heidelberg, Berlin, Heidelberg, 2003.
- ³ Yinan Zhang, Fei Wang, Jie Chao, Mo Xie, Huajie Liu, Muchen Pan, Enzo Kopperger, Xiaoguo Liu, Qian Li, Jiye Shi, Lianhui Wang, Friedrich Simmel, and Chunhai Fan. DNA origami cryptography for secure communication. Nat. Commun., 10:5469, 2019.
- ⁴ Mireille Bousquet-Mélou, AJ Guttmann, WP Orrick, and A Rechnitzer. Inversion relations, reciprocity and polyominoes. Ann. Comb., 3:223–249, 1999.
- ⁵ Floris AS Engelhardt, Florian Praetorius, Christian H Wachauf, Gereon Brüggenthies, Fabian Kohler, Benjamin Kick, Karoline L Kadletz, Phuong Nhi Pham, Karl L Behler, Thomas Gerling, et al. Custom-size, functional, and durable DNA origami with design-specific scaffolds. ACS Nano, 13(5):5015–5027, 2019.
- ⁶ Ralf Jungmann, Maier S Avendaño, Johannes B Woehrstein, Mingjie Dai, William M Shih, and Peng Yin. Multiplexed 3D cellular super-resolution imaging with DNA-PAINT and Exchange-PAINT. Nat. Methods, 11(3):313–318, 2014.
- ⁷ Gregor Posnjak, Xin Yin, Paul Butler, Oliver Bienek, Mihir Dass, Seungwoo Lee, Ian D Sharp, and Tim Liedl. Diamond-lattice photonic crystals assembled from DNA origami. Science, 384(6697):781–785, 2024.
- ⁸ Hendrik Dietz, Shawn M Douglas, and William M Shih. Folding DNA into twisted and curved nanoscale shapes. Science, 325(5941):725–730, 2009.
- ⁹ Fei Zhang, Shuoxing Jiang, Siyu Wu, Yulin Li, Chengde Mao, Yan Liu, and Hao Yan. Complex wireframe DNA origami nanostructures with multi-arm junction vertices. Nat. Nanotechnol., 10(9):779–784, 2015.
- ¹⁰ Shawn M Douglas, Hendrik Dietz, Tim Liedl, Björn Högberg, Franziska Graf, and William M Shih. Self-assembly of DNA into nanoscale three-dimensional shapes. Nature, 459(7245):414–418, 2009.
- ¹¹ Dongran Han, Suchetan Pal, Jeanette Nangreave, Zhengtao Deng, Yan Liu, and Hao Yan. DNA origami with complex curvatures in three-dimensional space. Science, 332(6027):342–346, 2011.
- ¹² Rémi Veneziano, Sakul Ratanalert, Kaiming Zhang, Fei Zhang, Hao Yan, Wah Chiu, and Mark Bathe. Designer nanoscale DNA assemblies programmed from the top down. Science, 352(6293):1534, 2016.
- ¹³ Susanne CM Reinhardt, Luciano A Masullo, Isabelle Baudrexel, Philipp R Steen, Rafal Kowalewski, Alexandra S Eklund, Sebastian Strauss, Eduard M Unterauer, Thomas Schlichthaerle, Maximilian T Strauss, et al. Ångström-resolution fluorescence microscopy. Nature, 617(7962):711–716, 2023.
- ¹⁴ Alexander Auer, Thomas Schlichthaerle, Johannes B Woehrstein, Florian Schueder, Maximilian T Strauss, Heinrich Grabmayr, and Ralf Jungmann. Nanometer-scale multiplexed super-resolution imaging with an economic 3D-DNA-PAINT microscope. ChemPhysChem, 19(22):3024–3034, 2018.
- ¹⁵ George D Dickinson, Golam Md Mortuza, William Clay, Luca Piantanida, Christopher M Green, Chad Watson, Eric J Hayden, Tim Andersen, Wan Kuang, Elton Graunard, et al. An alternative approach to nucleic acid memory. Nat. Commun., 12(1):2371, 2021.

- ¹⁶ TylerMSFT. Data Type Ranges — learn.microsoft.com. <https://learn.microsoft.com/en-us/cpp/cpp/data-type-ranges>, 2023. [Accessed 28-10-2024].
- ¹⁷ Joerg Schnitzbauer, Maximilian T Strauss, Thomas Schlichthaerle, Florian Schueder, and Ralf Jungmann. Super-resolution microscopy with DNA-paint. *Nat. Protoc.*, 12(6):1198–1228, 2017.
- ¹⁸ Mingjie Dai, Ralf Jungmann, and Peng Yin. Optical imaging of individual biomolecules in densely packed clusters. *Nat. Nanotechnol.*, 11(9):798–807, 2016.
- ¹⁹ Maximilian T Strauss, Florian Schueder, Daniel Haas, Philipp C Nickels, and Ralf Jungmann. Quantifying absolute addressability in DNA origami with molecular resolution. *Nat. Commun.*, 9(1):1600, 2018.
- ²⁰ Michael Riordan and Lillian Hoddeson. *Crystal fire: The birth of the information age*. WW Norton & Company, 1997.
- ²¹ Oded Goldreich. *Modern cryptography, probabilistic proofs and pseudorandomness*, volume 17. Springer Science & Business Media, 1998.
- ²² National Institute of Standards and Technology. Advanced encryption standard (AES). Technical report, Gaithersburg, MD, November 2001.
- ²³ Jeffrey Voas, Nir Kshetri, and Joanna F DeFranco. Scarcity and global insecurity: the semiconductor shortage. *IT Prof.*, 23(5):78–82, 2021.
- ²⁴ Arman Shehabi, Sarah Smith, Dale Sartor, Richard Brown, Magnus Herrlin, Jonathan Koomey, Eric Masanet, Nathaniel Horner, Inês Azevedo, and William Lintner. United states data center energy usage report. 6 2016.
- ²⁵ Karishma Matange, James M Tuck, and Albert J Keung. DNA stability: a central design consideration for DNA data storage systems. *Nat. Commun.*, 12(1):1358, 2021.
- ²⁶ Nick Goldman, Paul Bertone, Siyuan Chen, Christophe Dessimoz, Emily M LeProust, Botond Sipos, and Ewan Birney. Towards practical, high-capacity, low-maintenance information storage in synthesized DNA. *Nature*, 494:77–80, 2013.
- ²⁷ Andy Extance. How DNA could store all the world’s data. *Nature*, 537, 2016.
- ²⁸ Jonathan PL Cox. Long-term data storage in DNA. *Trends Biotechnol.*, 19(7):247–250, 2001.
- ²⁹ Qinghua Liu, Liman Wang, Anthony G Frutos, Anne E Condon, Robert M Corn, and Lloyd M Smith. DNA computing on surfaces. *Nature*, 403(13):175–178, 2000.
- ³⁰ Leonard M Adleman. Computing with DNA. *Sci. Am.*, 279(2):54–61, 1998.
- ³¹ Ravinderjit S Braich, Nickolas Chelyapov, Cliff Johnson, Paul WK Rothemund, and Leonard Adleman. Solution of a 20-variable 3-SAT problem on a DNA computer. *Science*, 296(5567):499–502, 2002.
- ³² Gheorghe Păun, Grzegorz Rozenberg, and Arto Salomaa. *DNA computing: new computing paradigms*. Springer, 1998.
- ³³ Lulu Qian, Erik Winfree, and Jehoshua Bruck. Neural network computation with DNA strand displacement cascades. *Nature*, 475(7356):368–372, 2011.
- ³⁴ Lulu Qian and Erik Winfree. Scaling up digital circuit computation with DNA strand displacement cascades. *Science*, 332(6034):1196–1201, 2011.
- ³⁵ Georg Seelig, David Soloveichik, David Yu Zhang, and Erik Winfree. Enzyme-free nucleic acid logic circuits. *Science*, 314(5805):1585–1588, 2006.
- ³⁶ George M Church, Yuan Gao, and Sriram Kosuri. Next-generation digital information storage in DNA. *Science*, 337(6102):1628, 2012.
- ³⁷ Luis Ceze, Jeff Nivala, and Karin Strauss. Molecular digital data storage using DNA. *Nat. Rev. Genet.*, 20(8):456–466, 2019.

- ³⁸ Lee Organick, Siena Dumas Ang, Yuan-Jyue Chen, Randolph Lopez, Sergey Yekhanin, Konstantin Makarychev, Miklos Z Racz, Govinda Kamath, Parikshit Gopalan, Bichlien Nguyen, et al. Random access in large-scale DNA data storage. *Nat. Biotechnol.*, 36(3):242–248, 2018.
- ³⁹ Paul K Hansma, Georg Schitter, Georg E Fantner, and Craig Prater. High-speed atomic force microscopy. *Science*, 314(5799):601–602, 2006.
- ⁴⁰ Anand Gole and Catherine J Murphy. Biotin- streptavidin-induced aggregation of gold nanorods: tuning rod- rod orientation. *Langmuir*, 21(23):10756–10762, 2005.
- ⁴¹ Kadir Aslan, Claudia C Luhrs, and Víctor H Pérez-Luna. Controlled and reversible aggregation of biotinylated gold nanoparticles with streptavidin. *J. Phys. Chem. B*, 108(40):15631–15639, 2004.
- ⁴² Andrea Doricchi, Casey M Platnich, Andreas Gimpel, Friederikee Horn, Max Earle, German Lanzavecchia, Aitziber L Cortajarena, Luis M Liz-Marzán, Na Liu, Reinhard Heckel, et al. Emerging approaches to DNA data storage: challenges and prospects. *ACS Nano*, 16(11):17552–17571, 2022.
- ⁴³ Kaikai Chen, Jinglin Kong, Jinbo Zhu, Niklas Ermann, Paul Predki, and Ulrich F Keyser. Digital data storage using DNA nanostructures and solid-state nanopores. *Nano Lett.*, 19(2):1210–1215, 2018.
- ⁴⁴ Kaikai Chen, Adnan Choudhary, Sarah E Sandler, Christopher Maffeo, Caterina Ducati, Aleksei Aksimentiev, and Ulrich F Keyser. Super-resolution detection of DNA nanostructures using a nanopore. *Adv. Mater.*, 35(12):2207434, 2023.
- ⁴⁵ Jinbo Zhu, Niklas Ermann, Kaikai Chen, and Ulrich F Keyser. Image encoding using multi-level DNA barcodes with nanopore readout. *Small*, 17(28):2100711, 2021.
- ⁴⁶ Sebastian Strauss and Ralf Jungmann. Up to 100-fold speed-up and multiplexing in optimized DNA-paint. *Nat. Methods*, 17(8):789–791, 2020.

Dear Editors and Reviewers,

We thank the Reviewers for their insights in improving our study and respectfully submit our responses to their comments on our manuscript entitled “*High-speed 3D DNA PAINT and unsupervised clustering to unlock 3D DNA origami cryptography*” (MS # NCOMMS-24-40556A). The Reviewers’ comments are written in blue, while the author’s responses to the comments are provided in black. The modifications introduced in the revised manuscript are highlighted in gray-shaded areas.

We appreciate all Reviewers’ support as well as thoughtful comments.

1 Reviewer #1 (Remarks to the Author)

The manuscript reports the first practical demonstration of 3D DNA-origami cryptography read out by high-speed 3D DNA-PAINT with unsupervised clustering, including controls and a clear analysis of detection efficiency and data-retrieval accuracy ($\approx 70\text{--}89\%$). Noteworthy advances include extending earlier 2D encryption schemes (e.g., Zhang 2019; Dickinson 2021) into 3D, showing that structural rigidity (tunnel and tetrapod designs) restores expected spatial readouts, and articulating the cryptographic role of scaffold-routing key space with discussion of multi-orthogonal/custom scaffolds. The added treatment of physical limits (evanescent depth, capacity estimates) and the sequencing/attack considerations strengthens the work’s framing. Methods, figures, and supplemental materials now provide sufficient detail for reproduction (sequences, buffers, imaging conditions, analysis workflow), and the methodology and analysis are sound. The main remaining weakness is practical scalability: per-origami payload is still demonstrated at ~ 2 bytes (with simulated paths to 4 bytes/multicolor) and high-throughput readout remains a bottleneck; a concise quantitative comparison table (AFM vs DNA-PAINT vs nanopore throughput/resolution) and release of analysis code/parameters and design files would further improve transparency. Overall, the data support the central claims, the contribution is significant to DNA nanotechnology, molecular cryptography, and super-resolution readout, and I recommend publication pending minor revisions to strengthen benchmarking and data/software availability.

We thank the reviewer for their thorough and insightful evaluation and for recognizing the significance of our demonstration of 3D DNA-origami cryptography and the methodological advances presented. We appreciate their constructive suggestions for improving benchmarking and data transparency.

We agree that benchmarking against other readout modalities would be valuable for contextualizing the throughput and resolution of 3D DNA-PAINT cryptographic readout. We have now added Supplementary Table 11 in the Supplementary Information, which summarizes key parameters (spatial resolution, per-origami readout rate, multiplexing potential, and compatibility with 3D structures) across AFM, DNA-PAINT, and nanopore-based approaches. This table highlights the complementary strengths and current throughput limitations of our method

Regarding the analysis code and design files, we have shared the codes which can be found in https://github.com/Jonathanzhao02/smlm_classification2d and <https://github.com/gwisna/DNA-origami-cryptography-code-and-data> and the source data presented in the main article and Supplementary Information which can be accessed through <https://doi.org/10.5281/zenodo.17362995>.

In summary, we have added a benchmarking table and provided public access to analysis and design resources. We believe these additions directly address the reviewer's suggestions and further enhance the transparency and reproducibility of our study. We added a sentence to discuss the comparison in the discussion:

Furthermore, we compared the nanoscale readout modalities of AFM, DNA-PAINT, and nanopore for DNA nanostructure-based data storage where DNA-PAINT excels in dimensional capability and resolution with comparable throughput to nanopore (Supplementary Table 11)

2 Reviewer #2 (Remarks to the Author)

The authors have sufficiently addressed my concerns, and the work is now suitable for publication in Nature Communications.

We thank the reviewer for the time and thoughtful comments on the revision that we have done and for accepting the manuscript to be published by Nature Communications. We are pleased to have addressed your concerns.

3 Reviewer #2 (Remarks on code availability)

I did not attempt to run the code, but the README file is highly instructive and the majority of the code is a replication of prior work.

We thank the reviewer on the code availability.

4 Reviewer #3 (Remarks to the Author)

The authors improved the manuscript and addressed the comments. The challenge for fast readout remains due to the clearly stated need for many imaging cycles. While practical applications might be challenging - the concept is interesting and the high-resolution readout make this suitable for NComms.

We thank the reviewer for the time and insightful comments on the revised manuscript and for accepting the manuscript to be published by Nature Communications.

5 Reviewer #4 (Remarks to the Author)

We thank the reviewer for the time and comments on the revised manuscript and for accepting the manuscript to be published by Nature Communications.

We thank the Reviewers for their appreciation of our work.

End of Responses to Referees

Thank you for your careful evaluation,

Gde Bimananda Mahardika Wisna, Prathamesh Chopade, and Rizal F. Hariadi